# Online Adaptation to Label Distribution Shift

**Ruihan Wu**
Cornell University
rw565@cornell.edu

**Chuan Guo**
Facebook AI Research
chuanguo@fb.com

**Yi Su**    **Kilian Q. Weinberger**
Cornell University
{ys756, kqw4@cornell.edu}

## Abstract

Machine learning models often encounter distribution shifts when deployed in the real world. In this paper, we focus on adaptation to label distribution shift in the *online setting*, where the test-time label distribution is continually changing and the model must dynamically adapt to it *without observing the true label*. Leveraging a novel analysis, we show that the lack of true label does not hinder estimation of the expected test loss, which enables the reduction of online label shift adaptation to conventional online learning. Informed by this observation, we propose adaptation algorithms inspired by classical online learning techniques such as Follow The Leader (FTL) and Online Gradient Descent (OGD) and derive their regret bounds. We empirically verify our findings under both simulated and real world label distribution shifts and show that OGD is particularly effective and robust to a variety of challenging label shift scenarios.

## 1 Introduction

A common assumption in machine learning is that the training set and test set are drawn from the same distribution [25]. However, this assumption often does not hold in practice when models are deployed in the real world [3, 28]. One common type of distribution shift is *label shift*, where the conditional distribution $p(\mathbf{x}|y)$ is fixed but the label distribution $p(y)$ changes over time. This phenomenon is most typical when the label $y$ is the causal variable and the feature $\mathbf{x}$ is the observation [31]. For instance, a model trained to diagnose malaria can encounter a much higher prevalence of the disease in tropical regions.

Prior work have primarily studied the problem of label shift in the offline setting [2, 4, 22, 42], where the phenomenon occurs only once after the model is trained. However, in many common scenarios, the label distribution can change continually over time. For example, the prevalence of a disease such as influenza changes depending on the season and whether an outbreak occurs. The distribution of news article categories is influenced by real world events such as political tension and the economy. In these scenarios, modeling the test-time label distribution as being stationary can be inaccurate and over-simplistic, especially if the model is deployed over a long period of time.

To address this shortcoming, we define and study the problem of *online label shift*, where the distribution shift is modeled as an online process. An adaptation algorithm in this setting is tasked with making sequential predictions on random samples from a drifting test distribution and dynamically adjusting the model's prediction in real-time. Different from online learning, the test label is *not* observed after making a prediction, hence making the problem much more challenging.

Nevertheless, we show that it is possible to adapt to the drifting test distribution in an online fashion, despite never observing a single label. In detail, we describe a method of obtaining unbiased estimates of the expected 0-1 loss and its gradient using only unlabeled samples. This allows the reduction of online label shift adaptation to conventional online learning, which we then utilize to define two algorithms inspired by classical techniques—Online Gradient Descent (OGD) and Follow The History (FTH)—the latter being a close relative of the Follow The Leader algorithm. Under

35th Conference on Neural Information Processing Systems (NeurIPS 2021).

**Framework 1** The general framework for online label shift adaptation.

**Input:** $\mathcal{A}$

1: $f_1 = f_0$;
2: **for** $t = 1, \cdots, T$ **do**
3:     Nature provides $\mathbf{x}_t$, where $(\mathbf{x}_t, y_t) \sim Q_t$ and $Q_t(\mathbf{x}|y_t) = Q_0(\mathbf{x}|y_t)$
4:     Learner predicts $f_t(\mathbf{x}_t)$
5:     Learner updates the classifier: $f_{t+1} = \mathcal{A}(f_0, \{\mathbf{x}_1, \cdots, \mathbf{x}_t\})$
6: **end for**

mild and empirically verifiable assumptions, we prove that OGD and FTH are as optimal as a fixed classifier that had knowledge of the shifting test distribution in advance.

To validate our theoretical findings, we evaluate our adaptation algorithms on CIFAR-10 [20] under simulated online label shifts, as well as on the ArXiv dataset[1] for paper categorization, which exhibits real world label shift across years of submission history. We find that OGD and FTH are able to consistently match or outperform the optimal fixed classifier, corroborating our theoretical results. Furthermore, OGD empirically achieves the best classification accuracy on average against a diverse set of challenging label shift scenarios, and can be easily adopted in practical settings.

## 2 Problem Setting

We first introduce the necessary notations and the general problem of label shift adaptation. Consider a classification problem with feature domain $\mathcal{X}$ and label space $\mathcal{Y} = \{1, \ldots, M\}$. We assume that a classifier $f : \mathcal{X} \rightarrow \mathcal{Y}$ operates by predicting a probability vector $P_f(\mathbf{x}) \in \Delta^{M-1}$, where $\Delta^{M-1}$ denotes the $(M-1)$-dimensional probability simplex. The corresponding classification function is $f(\mathbf{x}) = \arg\max_{y \in \mathcal{Y}} P_f(\mathbf{x})[y]$. While the focus of our paper is on neural networks, other models such as SVMs [7] and decision trees [27] that do not explicitly output probabilities can be calibrated to do so as well [26, 41].

**Label distribution shift.** Let $Q$ be a data distribution defined on $\mathcal{X} \times \mathcal{Y}$ and denote by $Q(\mathbf{x}|y)$ the class-conditional distribution, and by $Q(y)$ the marginal distribution, so that $Q(\mathbf{x}, y) = Q(\mathbf{x}|y)Q(y)$. Standard supervised learning assumes that the training distribution $Q_{\text{train}}$ and the test distribution $Q_{\text{test}}$ are identical. In reality, a deployed model often encounters *distributional shifts*, where the test distribution $Q_{\text{test}}$ may be substantially different from $Q_{\text{train}}$. We are interested in the scenario where the class-conditional distribution remains constant (*i.e.*, $Q_{\text{test}}(\mathbf{x}|y) = Q_{\text{train}}(\mathbf{x}|y)$ for all $\mathbf{x} \in \mathcal{X}, y \in \mathcal{Y}$) while the marginal label distribution changes (*i.e.*, $Q_{\text{test}}(y) \neq Q_{\text{train}}(y)$ for some $y \in \mathcal{Y}$), a setting we refer to as *label shift*. The problem of label shift adaptation is to design algorithms that can adjust the prediction of a classifier $f$ trained on $Q_{\text{train}}$ to perform well on $Q_{\text{test}}$.

**Offline label shift.** The general setting in this paper and in prior work on label shift adaptation is that the label marginal distribution $Q_{\text{test}}(y)$ is unknown. Indeed, the model may be deployed in a foreign environment where prior knowledge about the label marginal distribution is limited or inaccurate. Prior work tackle this challenge by estimating $Q_{\text{test}}(y)$ using *unlabeled samples* drawn from $Q_{\text{test}}$ [2, 4, 10, 22, 30]. Such adaptation methods operate under the *offline setting* since the distribution shift occurs only once, and the adaptation algorithm is provided with an unlabeled set of samples from the test distribution $Q_{\text{test}}$ upfront.

**Online label shift.** In certain scenarios, offline label shift adaptation may not be applicable. For example, consider a medical diagnosis model classifying whether a patient suffers from flu or hay fever. Although the two diseases share similar symptoms throughout the year, one is far more prevalent than the other, depending on the season and whether an outbreak occurs. Importantly, this label distribution shift is gradual and continual, and the model must make predictions on incoming patients in real-time without observing an unlabeled batch of examples first. Motivated by these challenging use cases, we deviate from prior work and study the problem where $Q_{\text{test}}$ is not stationary during test time and must be adapted towards in an online fashion.

Formally, we consider a discrete time scale with a shifting label distribution $Q_t(y)$ for $t = 0, 1, 2, \ldots$, where $Q_0 = Q_{\text{train}}$ denotes the training distribution. For each $t$, denote by $\mathbf{q}_t \in \Delta^{M-1}$

---

[1] https://www.kaggle.com/Cornell-University/arxiv

the label marginal probabilities so that $\mathbf{q}_t[i] = \mathbb{P}_{Q_t}(y_t = i)$ for all $i = 1, \ldots, M$. Let $\mathcal{H}$ be a fixed hypothesis space, let $f_0 \in \mathcal{H}$ be a classifier trained on samples from $Q_0$, and let $f_1 = f_0$.

At time step $t$, nature provides $(\mathbf{x}_t, y_t) \sim Q_t$ and the learner predicts the label of $\mathbf{x}_t$ using $f_t$. Similar to prior work on offline label shift, we impose a crucial but realistic restriction that *the true label $y_t$ and any incurred loss are both unobserved.* This restriction mimics real world situations such as medical diagnosis where the model may not receive any feedback while deployed. Despite this limitation, the learner may still seek to adapt the classifier to the test-time distribution after predicting on $\mathbf{x}_t$. Namely, the adaptation algorithm $\mathcal{A}$ takes as input $f_0$ and the unlabeled set of historical data $\{\mathbf{x}_1, \cdots, \mathbf{x}_t\}$ and outputs a new classifier $f_{t+1} \in \mathcal{H}$ for time step $t + 1$. This process is repeated until some end time $T$ and summarized in pseudo-code in Framework 1.

To quantify the effectiveness of the adaptation algorithm $\mathcal{A}$, we measure the expected regret across time steps $t = 1, \ldots, T$. Formally, we consider the expected 0-1 loss

$$\ell(f; Q) = \mathbb{P}_{(\mathbf{x},y)\sim Q}(f(\mathbf{x}) \neq y), \tag{1}$$

which we average across $t$ and measure against the best-in-class predictor from $\mathcal{H}$:

$$\text{Regret} = \frac{1}{T}\sum_{t=1}^{T}\ell(f_t; Q_t) - \inf_{f \in \mathcal{H}}\frac{1}{T}\sum_{t=1}^{T}\ell(f; Q_t). \tag{2}$$

The goal of online label shift adaptation is to design an algorithm $\mathcal{A}$ that minimizes expected regret.

## 3 Reduction to Online Learning

One of the main differences between online label shift adaptation and conventional online learning is that in the former, the learner *does not* receive any feedback after making a prediction. This restriction prohibits the application of classical online learning techniques that operate on a loss function observed after making a prediction at each time step. In this section, we show that in fact the expected 0-1 loss can be estimated using only the *unlabeled sample $\mathbf{x}_t$*, which in turn reduces the problem of online label shift adaptation to online learning. This reduction enables a variety of solutions derived from classical techniques, which we then analyze in section 4.

**Estimating the expected 0-1 loss.** For any classifier $f$ and distribution $Q_t$, let $C_{f,Q_t} \in \mathbb{R}^{M \times M}$ denote the confusion matrix for classifying samples from $Q_t$. Formally, $C_{f,Q_t}[i,j] = \mathbb{P}_{\mathbf{x}_t \sim Q_t(\cdot|y_t=i)}(f(\mathbf{x}_t) = j)$. Then the expected 0-1 loss of $f$ under distribution $Q_t$ (*cf.* Equation 1) can be written in terms of the confusion matrix $C_{f,Q_t}$ and the label marginal probabilities $\mathbf{q}_t$ by:

$$\ell(f; Q_t) = \mathbb{P}_{(\mathbf{x}_t,y_t)\sim Q_t}(f(\mathbf{x}_t) \neq y_t) = \sum_{i=1}^{M}\mathbb{P}_{\mathbf{x}_t \sim Q_t(\cdot|y_t=i)}(f(\mathbf{x}_t) \neq i) \cdot \mathbb{P}_{Q_t}(y_t = i)$$

$$= \sum_{i=1}^{M}\left(1 - \mathbb{P}_{\mathbf{x}_t \sim Q_t(\cdot|y=i)}(f(\mathbf{x}_t) = i)\right) \cdot \mathbf{q}_t[i] = \langle \mathbf{1} - \text{diag}\left(C_{f,Q_t}\right), \mathbf{q}_t \rangle,$$

where $\mathbf{1}$ denotes the all-1 vector, and $\text{diag}(C_{f,Q_t})$ is the diagonal of the confusion matrix. The difficulty in computing $\ell(f; Q_t)$ is that both $C_{f,Q_t}$ and $\mathbf{q}_t$ depend on the unknown distribution $Q_t$. However, by the label shift assumption that the conditional distribution $Q_t(\mathbf{x}|y)$ does not change over time, it immediately follows that $C_{f,Q_t} = C_{f,Q_0}$ $\forall t$, and therefore $\ell(f; Q_t)$ is *linear* in $\mathbf{q}_t$:

$$\ell(f; Q_t) = \langle \mathbf{1} - \text{diag}\left(C_{f,Q_0}\right), \mathbf{q}_t \rangle. \tag{3}$$

In the theorem below, we utilize this property to derive unbiased estimates of $\ell(f; Q_t)$ and its gradient $\nabla_f \ell(f; Q_t)$ with the assumption that $\text{diag}\left(C_{f,Q_0}\right)$ is differentiable with respect to $f$; we discuss this assumption in more detail in Section 4.1. The proof is included in the appendix and is inspired by prior work on offline label shift adaptation [22].

**Assumption 1.** $\text{diag}\left(C_{f,Q_0}\right)$ *is differentiable with respect to $f$.*

**Theorem 1.** *Let $f$ be any classifier and let $f_0$ be the classifier trained on data from $Q_0$. Suppose that $f_0$ predicts $f_0(\mathbf{x}_t) = i$ on input $\mathbf{x}_t \sim Q_t$ and let $\mathbf{e}_i$ denote the one-hot vector whose non-zero entry is $i$. If the confusion matrix $C_{f_0,Q_0}$ is invertible then $\hat{\mathbf{q}}_t = \left(C_{f_0,Q_0}^{\top}\right)^{-1}\mathbf{e}_i$ is an unbiased*

---
**Framework 2** The re-weighting framework for online label shifting adaptation.
---
**Input:** $\mathcal{A}, f_0, \mathbf{q}_0, D_0$

1: $f_1 = f_0$;
2: **for** $t = 1, \cdots, T$ **do**
3:     Nature provides $\mathbf{x}_t$, where $(\mathbf{x}_t, y_t) \sim Q_t$ and $Q_t(\mathbf{x}|y_t) = Q_0(\mathbf{x}|y_t)$;
4:     Learner predicts $f_t(\mathbf{x}_t)$
5:     Learner updates the re-weighting vector: $\mathbf{p}_{t+1} = \mathcal{A}(f_0, \mathbf{q}_0, D_0, \{\mathbf{x}_1, \cdots, \mathbf{x}_t\})$
6:     Learner updates the classifier: $f_{t+1} = g(\mathbf{x}; f_0, \mathbf{q}_0, \mathbf{p}_{t+1})$.
7: **end for**
---

*estimator of the label marginal probability vector* $\mathbf{q}_t$. *Further, we obtain unbiased estimators of the loss and gradient of $f$ for* $Q_t$ *with Assumption 1:*

$$\ell(f; Q_t) = \mathbb{E}_{Q_t}\left[\langle \mathbf{1} - \mathrm{diag}\left(C_{f,Q_0}\right), \hat{\mathbf{q}}_t \rangle\right],$$

$$\nabla_f \ell(f; Q_t) = \mathbb{E}_{Q_t}\left[J_f^\top \hat{\mathbf{q}}_t\right],$$

*where* $J_f = \frac{\partial}{\partial f}\left[\mathbf{1} - \mathrm{diag}\left(C_{f,Q_0}\right)\right]$ *denotes the Jacobian of* $\mathbf{1} - \mathrm{diag}\left(C_{f,Q_0}\right)$ *with respect to $f$.*

Note that in the theorem, the true confusion matrices $C_{f,Q_0}$ and $C_{f_0,Q_0}$ require knowledge of the training distribution $Q_0$ and are generally unobservable. We assume we have the full access of distribution $Q_0$ for the theoretical analysis in the remainder of the paper, including both values and gradients of $C_{f,Q_0}$ given any $f$. In practice, we can estimate the value of $C_{f,Q_0}$ by using a large labeled hold-out set $D_0$ drawn from $Q_0$ and the estimation error that can be reduced arbitrarily by increasing the size of $D_0$ [22]. For the detail of the practical gradient estimation, we discuss it in Section 4.1.

**Re-weighting algorithms for online label shift adaptation.** The result from Theorem 1 allows us to approximate the expected 0-1 loss and its gradient for *any* function $f$ over $Q_t$, which naturally inspires the following adaptation strategy: Choose a hypothesis space $\mathcal{G}$, and at each time step $t$, estimate the expected 0-1 loss and/or its gradient and apply any online learning algorithm to select the classifier $f_{t+1} \in \mathcal{G}$ for time step $t + 1$.

A natural choice for $\mathcal{G}$ is the hypothesis space of the original classifier $f_0$, *i.e.*, $\mathcal{G} = \mathcal{H}$. Indeed, the unbiased estimator for the gradient $\nabla_f \ell(f; Q_t)$ in Theorem 1 can be used to directly update $f_0$ with stochastic online gradient descent. However, as we assume that only the marginal distribution $Q_t(y)$ drifts and $Q_t(\mathbf{x}|y) = Q_0(\mathbf{x}|y)$, the conditional distribution $Q_t(y|\mathbf{x})$ must be a re-weighted version of $Q_0(y|\mathbf{x})$. Specifically,

$$Q_t(y|\mathbf{x}) = \frac{Q_t(y)}{Q_t(\mathbf{x})}Q_t(\mathbf{x}|y) = \frac{Q_t(y)}{Q_t(\mathbf{x})}Q_0(\mathbf{x}|y) = \frac{Q_t(y)}{Q_t(\mathbf{x})}\frac{Q_0(\mathbf{x})}{Q_0(y)}Q_0(y|\mathbf{x}) \propto \frac{Q_t(y)}{Q_0(y)}Q_0(y|\mathbf{x}). \quad (4)$$

Since the goal of $f_t$ is to approximate this conditional distribution, in principle, only a re-weighting of the predicted probabilities is needed to correct for any label distribution shift. A similar insight has been previously exploited to correct for training-time label imbalance [6, 8, 16, 39] and for offline label shift adaptation [4, 22]. We therefore focus our attention to the hypothesis space of re-weighted classifiers: $\mathcal{G}(f_0, \mathbf{q}_0) = \{g(\mathbf{x}; f_0, \mathbf{q}_0, \mathbf{p}) \mid \mathbf{p} \in \Delta^{M-1}\}$, where the classifier $g(\mathbf{x}; f_0, \mathbf{q}_0, \mathbf{p})$ is defined by the learned parameter vector $\mathbf{p}$ and takes the following form:

$$g(\mathbf{x}; f_0, \mathbf{q}_0, \mathbf{p}) = \arg\max_{y \in \mathcal{Y}} \frac{1}{Z(\mathbf{x})}\frac{\mathbf{p}[y]}{\mathbf{q}_0[y]}P_{f_0}(\mathbf{x})[y], \quad (5)$$

with $Z(\mathbf{x}) = \sum_{y \in \mathcal{Y}} \frac{\mathbf{q}_t[y]}{\mathbf{q}_0[y]}P_{f_0}(\mathbf{x})[y]$ being the normalization factor. We specialize the online label shift adaptation framework to the hypothesis space $\mathcal{G}(f_0, \mathbf{q}_0)$ in Framework 2, where the adaptation algorithm $\mathcal{A}$ focuses on generating a re-weighting vector $\mathbf{p}_t$ at each time step to construct the classifier $g(\mathbf{x}; f_0, \mathbf{q}_0, \mathbf{p}_t)$. The online learning objective can then be re-framed as choosing the re-weighting vector $\mathbf{p}_t$ that minimizes:

$$\text{Regret} = \frac{1}{T}\sum_{t=1}^{T}\ell(\mathbf{p}_t; \mathbf{q}_t) - \inf_{\mathbf{p} \in \Delta^{M-1}}\frac{1}{T}\sum_{t=1}^{T}\ell(\mathbf{p}; \mathbf{q}_t), \quad (6)$$

where $\ell(\mathbf{p}_t; \mathbf{q}_t) := \ell(g(\mathbf{x}; f_0, \mathbf{q}_0, \mathbf{p}_t); Q_t)$. For the remainder of this paper, we will analyze several classical online learning techniques under this framework.

# 4 Online Adaptation Algorithms

We now describe our main algorithms for online label shift adaptation. In particular, we present and analyze two online learning techniques—Online Gradient Descent (OGD) and Follow The History (FTH), the latter of which closely resembles Follow The Leader [32]. We show that under mild and empirically verifiable assumptions, OGD and FTH both achieve $\mathcal{O}(1/\sqrt{T})$ regret compared to the optimal fixed classifier in $\mathcal{G}(f_0, \mathbf{q}_0)$.

## 4.1 Algorithm 1: Online Gradient Descent

Online gradient descent (OGD) [32] is a classical online learning algorithm based on iteratively updating the hypothesis by following the gradient of the loss. Applied to our setting, the algorithm $\mathcal{A}_{\mathsf{ogd}}$ computes the stochastic gradient $\nabla_{\mathbf{p}}\ell(\mathbf{p}; \hat{\mathbf{q}}_t)\big|_{\mathbf{p}=\mathbf{p}_t} = J_{\mathbf{p}}(\mathbf{p}_t)^{\top}\hat{\mathbf{q}}_t$ using Theorem 1, where

$$J_{\mathbf{p}}(\mathbf{p}_t) = \frac{\partial}{\partial \mathbf{p}}\left(\mathbf{1} - \mathsf{diag}\left(C_{g(\cdot; f_0, \mathbf{q}_0, \mathbf{p}), Q_0}\right)\right)\bigg|_{\mathbf{p}=\mathbf{p}_t} \tag{7}$$

denotes the $M \times M$ Jacobian with respect to $\mathbf{p}$. OGD then applies the following update:

$$\mathbf{p}_{t+1} = \mathsf{Proj}_{\Delta^{M-1}}\left(\mathbf{p}_t - \eta \cdot \nabla_{\mathbf{p}}\ell(\mathbf{p}; \hat{\mathbf{q}}_t)|_{\mathbf{p}=\mathbf{p}_t}\right), \tag{8}$$

where $\eta > 0$ is the learning rate, and $\mathsf{Proj}_{\Delta^{M-1}}$ projects the updated vector onto $\Delta^{M-1}$.

The convergence rate of $\mathcal{A}_{\mathsf{ogd}}$ depends on properties of the loss function $\ell(\mathbf{p}; \mathbf{q})$. We empirically observe that $\ell(\mathbf{p}; \mathbf{q})$ is *approximately* convex in the re-weighting parameter vector $\mathbf{p}$, which we justify in detail in the appendix. To derive meaningful regret bounds for OGD, we further assume that the loss function $\ell(\mathbf{p}; \mathbf{q})$ is Lipschitz with respect to $\mathbf{p}$. Formally:

**Assumption 2** (Convexity). *$\forall \mathbf{q} \in \Delta^{M-1}$, $\ell(\mathbf{p}; \mathbf{q})$ is convex in $\mathbf{p}$.*

**Assumption 3** (Lipschitz-ness). *$\sup_{\mathbf{p} \in \Delta^{M-1}, i=1, \cdots, M} \left\| \nabla_{\mathbf{p}}\ell\left(\mathbf{p}; \left(C_{f_0, Q_0}^{\top}\right)^{-1}\mathbf{e}_i\right)\right\|_2$ is finite.*

Below, we provide our regret bound for OGD, which guarantees a $\mathcal{O}(1/\sqrt{T})$ convergence rate. The proof follows the classical proof for online gradient descent and is given in the appendix.

**Theorem 2** (Regret bound for OGD.). *Under Assumption 1, 2 and 3, let $L = \sup_{\mathbf{p} \in \Delta^{M-1}, i=1, \cdots, M} \left\| \nabla_{\mathbf{p}}\ell\left(\mathbf{p}; \left(C_{f_0, Q_0}^{\top}\right)^{-1}\mathbf{e}_i\right)\right\|_2$. If $\eta = \sqrt{\frac{2}{T}}\frac{1}{L}$ then $\mathcal{A}_{\mathsf{ogd}}$ satisfies:*

$$\mathbb{E}_{(\mathbf{x}_t, y_t) \sim Q_t}\left[\frac{1}{T}\sum_{t=1}^{T}\ell(\mathbf{p}_t; \mathbf{q}_t)\right] - \inf_{\mathbf{p} \in \Delta^{M-1}}\frac{1}{T}\sum_{t=1}^{T}\ell(\mathbf{p}; \mathbf{q}_t) \leq \sqrt{\frac{2}{T}}L.$$

**Gradient estimation.** Computing the unbiased gradient estimator $\nabla_{\mathbf{p}}\ell(\mathbf{p}; \hat{\mathbf{q}}_t)|_{\mathbf{p}=\mathbf{p}_t}$ involves the Jacobian term $J_{\mathbf{p}}(\mathbf{p}_t)$ in Equation 7, which is discontinous when estimated using the hold-out set $D_0$. More precisely, each entry of $\mathbf{1} - \mathsf{diag}\left(C_{g(\cdot; f_0, \mathbf{q}_0, \mathbf{p}), Q_0}\right)$ is the expected 0-1 loss for a particular class, whose estimate using $D_0$ is a step function. This means that taking the derivative naively will result in a gradient value of 0. To circumvent this issue, we apply finite difference approximation [1] for comput-

---

**Algorithm 3** Gradient estimator for $\nabla_{\mathbf{p}}\ell(\mathbf{p}; \mathbf{q})$

**Input:** $\mathbf{p}, \mathbf{q}, \delta, k$
1: **for** $i = 1, \ldots, M$ **do**
2: $\quad \Delta_i := \ell(\mathbf{p} + j\delta \cdot \mathbf{e}_i, \mathbf{q}) - \ell(\mathbf{p} - j\delta \cdot \mathbf{e}_i, \mathbf{q})$
3: $\quad \hat{\nabla}_{\mathbf{p}}[i] := \sum_{j=1}^{k}\alpha_j\Delta_i/(2\delta j)$, where $\alpha_j = 2 \cdot (-1)^{j+1}\binom{k}{k-j}/\binom{k+j}{k}$.
4: **end for**
5: **Return:** $\hat{\nabla}_{\mathbf{p}}$

---

ing $\nabla_{\mathbf{p}}\ell(\mathbf{p}; \hat{\mathbf{q}}_t)|_{\mathbf{p}=\mathbf{p}_t}$, which is detailed in Algorithm 3. We also apply smoothing to compute the average estimated gradient around the target point $\mathbf{p}_t$ to improve gradient stability.

Alternatively, we can minimize a smooth surrogate of the 0-1 loss that enables direct gradient computation. In detail, we define $\ell^{\mathsf{prob}}(f; Q) := \mathbb{E}_{(\mathbf{x}, y) \sim Q}[1 - P_f(\mathbf{x})[y]] \in [0, 1]$ so that $\ell^{\mathsf{prob}} = \ell$ when $P_f$ outputs one-hot probability vectors. Furthermore, we show that $\ell^{\mathsf{prob}}$ enjoys the same unbiased estimation properties as that of $\ell$ in Theorem 1, admits smooth gradient estimates using a finite

hold-out set $D_0$, and is classification-calibrated in the sense of Tewari and Bartlett [38]. The formal statement and proof of the above properties are given in the appendix. In section 5 we empirically evaluate OGD using both the finite difference approach and the surrogate loss approach for gradient estimation.

## 4.2 Algorithm 2: Follow The History

Next, we describe Follow The History (FTH)—a minor variant of the prominent online learning strategy known as Follow The Leader (FTL) [32]. In FTL, the basic intuition is that the predictor $f_t$ for time step $t$ is the one that minimizes the average loss from the previous $t-1$ time steps. Formally:

$$\mathbf{p}_{t+1} = \arg\min_{\mathbf{p} \in \Delta^{M-1}} \frac{1}{t} \sum_{\tau=1}^{t} \ell\left(\mathbf{p}; \hat{\mathbf{q}}_\tau\right) = \arg\min_{\mathbf{p} \in \Delta^{M-1}} \ell\left(\mathbf{p}; \frac{1}{t} \sum_{\tau=1}^{t} \hat{\mathbf{q}}_\tau\right), \tag{9}$$

where the second inequality holds by linearity of $\ell$. However, faithfully executing FTL requires optimizing the loss in Equation 9 at each time step, which could be very inefficient since multiple gradients of $\ell$ need to be computed as opposed to a single gradient computation for OGD.

To address this efficiency concern, observe that if the original classifier $f_0$ is Bayes optimal, then for any $\mathbf{q}_t \in \Delta^{M-1}$, the minimizer over the re-weighting vector $\mathbf{p}$ of $\ell(\mathbf{p}; \mathbf{q}_t)$ is the test-time label marginal probability vector $\mathbf{q}_t$ itself (*cf.* Equation 4). In fact, we show in the appendix that this assumption often holds *approximately* in practice, especially when $f_0$ achieves a low error on $Q_0$ and is well-calibrated [12, 26, 41]. Assuming this approximation error is bounded by some $\delta \geq 0$, we can derive a more efficient update rule and a corresponding regret bound. Formally:

**Assumption 4** (Symmetric optimality). *For any* $\mathbf{q} \in \Delta^{M-1}$, $\|\mathbf{q} - \arg\min_{\mathbf{p} \in \Delta^{M-1}} \ell(\mathbf{p}; \mathbf{q})\|_2 \leq \delta$.

We define the more efficient Follow The History update rule $\mathcal{A}^{\text{fth}}$ as: $\mathbf{p}_{t+1} = \frac{1}{t} \sum_{\tau=1}^{t} \hat{\mathbf{q}}_\tau$, which is a simple average of the estimates $\hat{\mathbf{q}}_\tau$ from all previous iterations of the algorithm. FTH coincides with FTL when Assumption 4 holds with $\delta = 0$. In the following theorem, we derive the regret bound for FTH when $\delta = 0$ but prove the general case of $\delta \geq 0$ in the appendix. The theorem relies on an assumption of Lipschitz-ness that is slightly different from Assumption 3. We formally state them as below:

**Assumption 5** (Lipschitz-ness for FTH). $\sup_{\mathbf{p},\mathbf{q} \in \Delta^{M-1}} \|\nabla_{\mathbf{p}} \ell(\mathbf{p}; \mathbf{q})\|_2$ *is finite.*

**Theorem 3.** *Under Assumption 4 and 5 with* $\delta = 0$, *with probability at least* $1 - 2MT^{-7}$ *over samples* $(\mathbf{x}_t, y_t) \sim Q_t$ *for* $t = 1, \ldots, T$ *we have that* $\mathcal{A}^{\text{fth}}$ *satisfies:*

$$\frac{1}{T} \sum_{t=1}^{T} \ell(\mathbf{p}_t; \mathbf{q}_t) - \inf_{\mathbf{p} \in \Delta^{M-1}} \sum_{t=1}^{T} \ell(\mathbf{p}; \mathbf{q}_t) \leq 2L\frac{\ln T}{T} + 4Lc\sqrt{\frac{M \ln T}{T}},$$

*where* $c = 2\max_{i=1,\ldots,M} \left\| \left(C_{f_0,Q_0}^{\top}\right)^{-1} \mathbf{e}_i \right\|_{\infty}$ *and* $L = \sup_{\mathbf{p},\mathbf{q} \in \Delta^{M-1}} \|\nabla_{\mathbf{p}} \ell(\mathbf{p}; \mathbf{q})\|_2$.

## 5 Experiment

In this section, we empirically evaluate the proposed algorithms on datasets with both simulated and real world online label shifts. Our simulated label shift experiment is performed on CIFAR-10 [20], where we vary the shift process and explore the robustness of different algorithms. For real world label shift, we evaluate on the ArXiv dataset[2] for paper categorization, where label shift occurs naturally over years of paper submission due to changing interest in different academic disciplines.

### 5.1 Experiment set-up

**Online algorithms set-up.** We evaluate both the OGD and FTH algorithms from section 4. For OGD, we use the learning rate $\eta = \sqrt{\frac{2}{T}\frac{1}{L}}$ suggested by Theorem 2, where $L$ is estimated by taking the maximum over $\{\mathbf{e}_y : y \in \mathcal{Y}\}$ for 100 vectors $\mathbf{p}'$ uniformly sampled from $\Delta^{M-1}$. The

---

[2] https://www.kaggle.com/Cornell-University/arxiv

gradient estimate can be derived using either the finite difference method in Algorithm 3 or by directly differentiating the surrogate loss $\ell^{\text{prob}}$. We evaluate both methods in our experiments.

For FTH, we evaluate both the algorithm $\mathcal{A}^{\text{fth}}$ defined in subsection 4.2 and a heuristic algorithm $\mathcal{A}^{\text{ftfwh}}$ which we call *Follow The Fixed Window History (FTFWH)*. Different from FTH where $\mathbf{p}_{t+1}$ is the simple average of $\hat{\mathbf{q}}_\tau$ across all previous time steps $\tau = 1, \ldots, t$, FTFWH averages across previous estimates $\hat{\mathbf{q}}_\tau$ in a fixed window of size $w$, *i.e.*, $\mathbf{p}_{t+1} = \sum_{\tau=\max\{1,t-w+1\}}^{t} \hat{\mathbf{q}}_\tau / \min\{w, t\}$. Intuitively, FTFWH assumes that the distribution $Q_t$ as fixed for $w$ time steps and solves the offline label adaptation problem for the next time step. We use three different window lengths $w = 100, 1000, 10000$ in our experiments. We will show that FTFWH can be optimal at times but is inconsistent in performance, especially when the window size $w$ coincides with the periodicity in the label distribution shift.

**Baselines.** In addition, we consider the following baseline classifiers as benchmarks against online adaptation algorithms.

- *Base Classifier (BC)* refers to the classifier $f_0$ without any online adaptation, which serves as a reference point for evaluating the performance of online adaptation algorithms.
- *Optimal Fixed Classifier (OFC)* refers to the best-in-class classifier in $\mathcal{G}(f_0, \mathbf{q}_0)$, which is the optimum in Equation 6. We denote the re-weighting vector that achieves this optimum as $\mathbf{p}^{\text{opt}}$. In simulated label shift, we can define the ground truth label marginal probability vector $\mathbf{q}_t$ and optimize for $\mathbf{p}^{\text{opt}}$ directly. For the experiment on ArXiv, we derive the optimum using the empirical loss: $\mathbf{p}^{\text{opt}} = \arg\min_{\mathbf{p} \in \Delta^{M-1}} \ell\left(\mathbf{p}; \frac{1}{T}\sum_{t=1}^{T} \mathbf{e}_{y_t}\right)$ where $y_t$ is the ground truth label at time $t$. Note that OFC is not a practical algorithm since $y_t$ is not observed, but it can be used to benchmark different adaptation algorithms and estimate their empirical regret.

**Evaluation metric.** Computing the actual regret requires access to $\mathbf{q}_t$ for all $t$, which we do not observe in the real world dataset. To make all evaluations consistent, we report the **average error** $\frac{1}{T}\sum_{t=1}^{T} \mathbb{1}\left(g(\mathbf{x}_t; f_0, \mathbf{q}_0, \mathbf{p}_t) \neq y_t\right)$ to approximate $\frac{1}{T}\sum_{t=1}^{T} \ell(\mathbf{p}_t; \mathbf{q}_t)$. This approximation is valid for large $T$ due to its exponential concentration rate by the Azuma–Hoeffding inequality.

## 5.2 Evaluation on CIFAR-10 under simulated shift

**Dataset and model.** We conduct our simulated label shift experiments on CIFAR-10 [20] with a ResNet-18 [13] classifier. We divide the original training set into train and validation by a ratio of $3:2$. The training set is used to train the base model $f_0$, and the validation set $D_0$ is used for both temperature scaling calibration [12] and to estimate the confusion matrix. The original test set is for the online data sampling. Additional training details are provided in the appendix.

**Simulated shift processes.** Let $\mathbf{q}^{(1)}, \mathbf{q}^{(2)} \in \Delta^{M-1}$ be two fixed probability vectors. We define the following simulated shift processes for the test-time label marginal probability vector $\mathbf{q}_t$.

- *Constant shift*: $\mathbf{q}_t = \mathbf{q}^{(1)}$ for all $t$, which coincides with the setting of offline label shift.
- *Monotone shift*: $\mathbf{q}_t$ interpolates from $\mathbf{q}^{(1)}$ to $\mathbf{q}^{(2)}$, *i.e.*, $\mathbf{q}_t := \left(1 - \frac{t}{T}\right)\mathbf{q}^{(1)} + \left(\frac{t}{T}\right)\mathbf{q}^{(2)}$.
- *Periodic shift*: $\mathbf{q}_t$ alternates between $\mathbf{q}^{(1)}$ and $\mathbf{q}^{(2)}$ at a fixed period of $T_p$. We test under three different periods $T_p = 100, 1000, 10000$.
- *Exponential periodic shift*: $\mathbf{q}_t$ alternates between $\mathbf{q}^{(1)}$ and $\mathbf{q}^{(2)}$ with an exponentially growing period. Formally, $\forall t \in [k^{2i}, k^{2i+1}]$, $\mathbf{q}_t := \mathbf{q}^{(1)}$; $\forall t \in [k^{2i-1}, k^{2i}]$, $\mathbf{q}_t := \mathbf{q}^{(2)}$. We use $k = 2, 5$ for our experiments.

In our experiments, $\mathbf{q}^{(1)}$ and $\mathbf{q}^{(2)}$ are defined to concentrate on the *dog* and *cat* classes, respectively. That is, $\mathbf{q}^{(1)}[\text{dog}] = 0.55$ and $\mathbf{q}^{(1)}[y] = 0.05$ for all other classes $y$, and similarly for $\mathbf{q}^{(2)}$. The end time $T$ is set to $100,000$ for all simulation experiments. All results are repeated using three different random seeds that randomize the samples drawn at each time step $t$.

**Results.** Table 1 shows the average error of various adaptation algorithms when applied to the simulated label shift processes. All adaptation algorithms can outperform the base classifier $f_0$ (except for FTFWH for periodic shift with $T_p = 100$), which serves as a sanity check that the algorithm is indeed adapting to the test distribution.

| Method | Simulated Label Shift | | | | | | |
|---|---|---|---|---|---|---|---|
| | Constant | Monotone | Periodic $T_p = 100$ | Periodic $T_p = 1000$ | Periodic $T_p = 10000$ | Exp. Periodic $k = 2$ | Exp. Periodic $k = 5$ |
| Base Classifier ($f_0$) | 12.43± 0.04 | 11.63± 0.08 | 11.63± 0.09 | 11.62± 0.08 | 11.63± 0.10 | 11.67± 0.07 | 11.81± 0.11 |
| Opt. Fixed Classifier | 7.78± 0.10 | 10.24± 0.08 | 10.25± 0.08 | 10.24± 0.08 | 10.24± 0.08 | 10.27± 0.07 | 10.25± 0.09 |
| FTH | 7.68± 0.11 | 10.36± 0.06 | 10.27± 0.08 | 10.25± 0.06 | 10.33± 0.10 | 10.23± 0.06 | 10.25± 0.04 |
| FTFWH $w = 10^2$ | 8.71± 0.10 | 10.19± 0.05 | 12.15± 0.01 | 8.85± 0.04 | 8.46± 0.11 | 8.52± 0.06 | 8.54± 0.05 |
| FTFWH $w = 10^3$ | 7.74± 0.07 | 9.52± 0.10 | 10.25± 0.07 | 11.16± 0.12 | 7.84± 0.09 | 7.77± 0.06 | 7.67± 0.08 |
| FTFWH $w = 10^4$ | 7.67± 0.08 | 9.53± 0.10 | 10.26± 0.07 | 10.26± 0.06 | 10.83± 0.07 | 8.93± 0.07 | 8.46± 0.07 |
| OGD (finite diff.) | 8.08± 0.08 | 9.79± 0.09 | 10.71± 0.10 | 10.62± 0.09 | 10.11± 0.06 | 8.99± 0.05 | 8.56± 0.12 |
| OGD (surrogate loss) | 7.78± 0.11 | 9.75± 0.07 | 10.24± 0.06 | 10.21± 0.07 | 10.05± 0.09 | 8.92± 0.04 | 8.50± 0.11 |

Table 1: Average error (%) for different adaptation algorithms under simulated label shift on CIFAR-10. Standard deviation is computed across three runs. Results that are better than the OFC benchmark by 0.5 or more are highlighted in blue, and results that are worse than the OFC benchmark by 0.5 or more are highlighted in red.

| Method | Base ($f_0$) | Opt. Fixed | FTH | FTFWH $w = 10^2$ | FTFWH $w = 10^3$ | FTFWH $w = 10^4$ | OGD finite diff. | OGD surr. loss |
|---|---|---|---|---|---|---|---|---|
| Avg. Error (%) | 27.21 | 25.56 | 25.62 | 30.14 | 26.09 | 25.87 | **25.52** | 25.70 |

Table 2: Average test error (%) on the ArXiv dataset.

Comparison with the optimal fixed classifier (OFC) reveals more insightful characteristics of the different algorithms and we discuss each algorithm separately as follows.

- The performance of *Follow The History (FTH)* is guaranteed to be competitive with *OFC* by Theorem 3 when the base model $f_0$ is well-calibrated. Indeed, as shown in the table, the average error for FTH is close to that of *OFC* for all shift processes. However, it is also very conservative and can never achieve better performance than *OFC* by a margin larger than 0.5.
- *Follow The Fixed Window History (FTFWH)* with a suitably chosen window size performs very well empirically, especially for constant shift, monotone shift, and exponential periodic shift. However, when encountering periodic shift with the periodicity $T_p$ equal to the window size $w$ (highlighted in red), FTFWH is consistently subpar compared to OFC, and sometimes even worse than the base classifier $f_0$. Since real world distribution shifts are often periodic in nature (*e.g.*, seasonal trends for flu and hay fever), this result suggests that deploying FTFWH may require knowledge of the periodicity in advance, which may not be feasible. In fact, we show in our experiment on real world label shift on ArXiv that FTFWH is never better than FTH and OGD.
- *Online gradient descent (OGD)* with learning rate $\eta = \sqrt{\frac{2}{T}} \frac{1}{L}$ is also guaranteed to be as good as *OFC* by Theorem 2, which is empirically observed as well. Moreover, unlike *FTH* which only achieves an average error no worse than that of *OFC*, *OGD* is able to outperform *OFC* on certain scenarios such as monotone shift, periodic shift with $T_p = 10000$ and exponential periodic shift with $K = 2, 5$. We also observe that OGD using the surrogate loss for gradient estimation is consistently better than OGD with finite difference gradient estimation.

Overall, we observe that *OGD* is the most reliable adaptation algorithm from the above simulation results, as it is uniformly as good as *OFC* and sometimes can achieve even better results than *OFC*.

## 5.3 Evaluation on ArXiv under real world distribution shift

**Dataset and model.** We experiment on the ArXiv dataset for categorization of papers from the Computer Science domain into 23 refined categories[3]. There are a total of $233,748$ papers spanning from the year 1991 to 2020, from which we sort by submission time and divide by a ratio of $2 : 1 : 1$ into the training, validation, and test sets. For each paper, we compute the tf-idf vector of its abstract as the feature $\mathbf{x}$, and we use the first category in its primary category set as the true label $y$. The base model $f_0$ is an $L_2$-regularized multinomial regressor. Same as in the simulated shift experiments, the validation set $D_0$ is used to calibrate the base model $f_0$ and estimate the confusion matrix. Additional details on data processing and training are given in the appendix.

---

[3]There are actually 40 categories in the Computer Science domain, from which we select the 23 most populated categories.

**Results.** Table 2 shows the average error for each adaptation algorithm over the test set, which consists of papers sorted by submission time with end time $T = 58,437$. In contrast to the simulated shift experiments in subsection 5.2, FTFWH is consistently worse than the optimal fixed classifier (OFC), especially for window size $w = 100$ where it is even worse than the base classifier $f_0$. This result shows that despite the good performance of FTFWH for simulated label shifts on CIFAR-10, it encounters significant challenges when deployed to the real world.

On the other hand, FTH and OGD both achieve an average error close to that of OFC, which again validates the theoretical regret bounds in Theorem 2 and Theorem 3. Given the empirical observation of OGD's performance on both simulated and real world label shift, as well as its conceptual simplicity and ease of implementation, we therefore recommend it as a practical go-to solution for online label shift adaptation in real world settings.

# 6   Related Work

Label shift adaptation has received much attention lately. The seminal work by Saerens et al. [30] made the critical observation that the label shift condition of $p(\mathbf{x}|y)$ being stationary implies the optimality of re-weighted classifiers. They defined an Expectation-Maximization (EM) procedure to utilize this insight and learn the re-weighting vector that maximizes likelihood on the test distribution. Alexandari et al. [2] studied this approach further and discovered that calibrating the model can lead to a significant improvement in performance. Another prominent strategy for learning the re-weighting vector is by inverting the confusion matrix [22], which inspired our approach for the online setting. Extensions to this method include regularizing the re-weighting vector [4], generalizing the label shift condition to feature space rather than input space [37], and unifying the confusion matrix approach and the EM approach [10].

Another type of test-time distribution shift that has been widely studied is *covariate shift*. Differing from the label shift assumption that $p(\mathbf{x}|y)$ is constant, covariate shift assumes instead that $p(y|\mathbf{x})$ is a constant and $p(\mathbf{x})$ changes between training and test distributions. Earlier work by Lin et al. [21] relied on the assumption that the density function $p(\mathbf{x})$ is known, while subsequent work relaxed this assumption by estimating the density from data [11, 17, 34, 40]. Online extensions to the covariate shift problem have also been considered. Solutions to this problem either rely on the knowledge of test labels [18], or use unsupervised test-time adaptation methods that are tailored to visual applications [14, 24, 36].

More broadly, both label shift adaptation and covariate shift adaptation can be categorized under the general problem of *domain adaptation*. This problem is much more challenging because a test sample may not necessarily belong to the support of the training distribution. In this setting, there is a large line of theoretical work that prove performance guarantees under bounded training and test distribution divergence [5, 15, 23, 33], as well as empirical methods for domain adaptation in the realm of deep learning [9, 19, 29, 35].

# 7   Conclusion

We presented a rigorous framework for studying online label shift adaptation, which addresses limitations in prior work on offline label shift. Under our framework, we showed that it is possible to obtain unbiased estimates of the expected 0-1 loss and its gradient without observing a single label at test time. This reduction enables the application of classical techniques from online learning to define practical adaptation algorithms. We showed that these algorithms admit rigorous theoretical guarantees on performance, while at the same time perform very well empirically and can adapt to a variety of challenging label shift scenarios.

One potential future work is to relax Assumption 2 to weak convexity assumption. With this relaxation, one needs to take a closer look to online learning techniques and applies it into the online label shift problem with the reduction introduced in this paper and potential additional reduction. Another extension is that we focused on re-weighting algorithms for online label shift adaptation in this paper, the reduction presented in section 3 technically enables a larger set of solutions. Indeed, one possible future direction is to directly apply OGD to update the base classifier $f_0$, which may further improve the adaptation algorithm's performance.

## Acknowledgments and Disclosure of Funding

RW and KQW are supported by grants from the National Science Foundation NSF (III-1618134, III-1526012, IIS-1149882, and IIS-1724282), the Bill and Melinda Gates Foundation, and the Cornell Center for Materials Research with funding from the NSF MRSEC program (DMR-1719875), and SAP America.

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
