# A   Proof in Section 4

**Theorem 1.** *Let $f$ be any classifier and let $f_0$ be the classifier trained on data from $Q_0$. Suppose that $f_0$ predicts $f_0(\mathbf{x}_t) = i$ on input $\mathbf{x}_t \sim Q_t$ and let $\mathbf{e}_i$ denote the one-hot vector whose non-zero entry is $i$. If the confusion matrix $C_{f_0, Q_0}$ is invertible then $\hat{\mathbf{q}}_t = \left( C_{f_0, Q_0}^\top \right)^{-1} \mathbf{e}_i$ is an unbiased estimator of the label marginal probability vector $\mathbf{q}_t$. Further, we obtain unbiased estimators of the loss and gradient of $f$ for $Q_t$ with Assumption 1:*

$$\ell(f; Q_t) = \mathbb{E}_{Q_t} \left[ \langle \mathbf{1} - \text{diag}\left( C_{f, Q_0} \right), \hat{\mathbf{q}}_t \rangle \right],$$

$$\nabla_f \ell(f; Q_t) = \mathbb{E}_{Q_t} \left[ J_f^\top \hat{\mathbf{q}}_t \right],$$

*where $J_f = \frac{\partial}{\partial f} \left[ \mathbf{1} - \text{diag}\left( C_{f, Q_0} \right) \right]$ denotes the Jacobian of $\mathbf{1} - \text{diag}\left( C_{f, Q_0} \right)$ with respect to $f$.*

*Proof of Theorem 1.* Let $\boldsymbol{\alpha}[i] = \mathbb{P}_{(\mathbf{x}_t, y_t) \sim Q_t}(f_0(\mathbf{x}_t) = i)$ for $i = 1, \dots, M$ be the proportion of samples drawn from $Q_t$ that the model $f_0$ predicts as class $i$. Then:

$$\boldsymbol{\alpha}[i] = \sum_{j=1}^M \mathbb{P}_{\mathbf{x}_t \sim Q_t(\cdot | y_t = j)}(f_0(\mathbf{x}_t) = i) \cdot \mathbf{q}_t[j] = C_{f_0, Q_t}[:, i]^\top \mathbf{q}_t, \tag{10}$$

hence $\boldsymbol{\alpha} = C_{f_0, Q_t}^\top \mathbf{q}_t$. Then if the confusion matrix $C_{f_0, Q_t} = C_{f_0, Q_0}$ is invertible, we can estimate $\mathbf{q}_t$ using $\hat{\mathbf{q}}_t = \left( C_{f_0, Q_0}^\top \right)^{-1} \mathbf{e}_i$, with $\hat{\mathbf{q}}_t$ satisfying:

$$\mathbb{E}[\hat{\mathbf{q}}_t] = \mathbb{E} \left[ \left( C_{f_0, Q_0}^\top \right)^{-1} \mathbf{e}_i \right] = \left( C_{f_0, Q_0}^\top \right)^{-1} \mathbb{E}[\mathbf{e}_i] = \left( C_{f_0, Q_0}^\top \right)^{-1} \boldsymbol{\alpha} = \mathbf{q}_t, \tag{11}$$

so $\hat{\mathbf{q}}_t$ is an unbiased estimator for $\mathbf{q}_t$. From Equation 3 and the fact that $Q_0$ is independent of $\mathbf{q}_t$, we have that $\langle \mathbf{1} - \text{diag}\left( C_{f, Q_0} \right), \hat{\mathbf{q}}_t \rangle$ and $J_f^\top \hat{\mathbf{q}}_t$ are unbiased estimators for $\ell(f; Q_t)$ and $\nabla_f \ell(f; Q_t)$, respectively. $\qquad \square$

**Theorem 2** (Regret bound for OGD). *Under Assumption 1, 2 and 3, let $L = \sup_{\mathbf{p} \in \Delta^{M-1}, i=1, \cdots, M} \left\| \nabla_\mathbf{p} \ell \left( \mathbf{p}; \left( C_{f_0, Q_0}^\top \right)^{-1} \mathbf{e}_i \right) \right\|_2$. If $\eta = \sqrt{\frac{2}{T} \frac{1}{L}}$ then $\mathcal{A}_{\text{ogd}}$ satisfies:*

$$\mathbb{E}_{(\mathbf{x}_t, y_t) \sim Q_t} \left[ \frac{1}{T} \sum_{t=1}^T \ell(\mathbf{p}_t; \mathbf{q}_t) \right] - \inf_{\mathbf{p} \in \Delta^{M-1}} \frac{1}{T} \sum_{t=1}^T \ell(\mathbf{p}; \mathbf{q}_t) \le \sqrt{\frac{2}{T}} L.$$

*Proof of Theorem 2.* Following the similar argument as Theorem 4.1 in [32], for any fixed $\mathbf{p}$,

$$\mathbb{E}_{(\mathbf{x}_t, y_t) \sim Q_t} \left[ \frac{1}{T} \sum_{t=1}^T \ell(\mathbf{p}_t, \mathbf{q}_t) \right] - \frac{1}{T} \sum_{t=1}^T \ell(\mathbf{p}, \mathbf{q}_t) = \mathbb{E} \left[ \frac{1}{T} \sum_{t=1}^T \ell(\mathbf{p}_t, \mathbf{q}_t) - \ell(\mathbf{p}, \mathbf{q}_t) \right]$$

$$\le \frac{1}{T} \mathbb{E} \left[ \sum_{t=1}^T (\mathbf{p}_t - \mathbf{p}) \cdot \nabla_\mathbf{p} \ell(\mathbf{p}_t, \mathbf{q}_t) \right]$$

$$= \frac{1}{T} \mathbb{E} \left[ \sum_{t=1}^T (\mathbf{p}_t - \mathbf{p}) \cdot \mathbb{E} \left[ \nabla_\mathbf{p} \ell(\mathbf{p}_t, \hat{\mathbf{q}}_t) | \mathbf{p}_t \right] \right]$$

$$= \frac{1}{T} \mathbb{E} \left[ \sum_{t=1}^T (\mathbf{p}_t - \mathbf{p}) \cdot \nabla_\mathbf{p} \ell(\mathbf{p}_t, \hat{\mathbf{q}}_t) \right],$$

where the second inequality holds by the law of total probability. To bound $(\mathbf{p}_t - \mathbf{p}) \cdot \nabla_\mathbf{p} \ell(\mathbf{p}_t, \hat{\mathbf{q}}_t)$,

$$\|\mathbf{p}_{t+1} - \mathbf{p}\|_2^2 = \|\text{Proj}_{\Delta^{M-1}} \left( \mathbf{p}_t - \eta \cdot \nabla_\mathbf{p} \ell(\mathbf{p}_t, \hat{\mathbf{q}}_t) \right) - \mathbf{p}\|_2^2$$

$$\le \|\mathbf{p}_t - \eta \cdot \nabla_\mathbf{p} \ell(\mathbf{p}_t, \hat{Q}_t) - \mathbf{p}\|_2^2$$

$$= \|\mathbf{p}_t - \mathbf{p}\|_2^2 + \eta^2 \|\nabla_\mathbf{p} \ell(\mathbf{p}_t, \hat{\mathbf{q}}_t)\|_2^2 - 2\eta(\mathbf{p}_t - \mathbf{p}) \cdot \nabla_\mathbf{p} \ell(\mathbf{p}_t, \hat{\mathbf{q}}_t),$$

which implies

$$(\mathbf{p}_t - \mathbf{p}) \cdot \nabla_{\mathbf{p}}\ell(\mathbf{p}_t, \hat{\mathbf{q}}_t) \leq \frac{1}{2\eta}\left(||\mathbf{p}_t - \mathbf{p}||_2^2 - ||\mathbf{p}_{t+1} - \mathbf{p}||_2^2\right) + \frac{\eta}{2}||\nabla_{\mathbf{p}}\ell(\mathbf{p}_t, \hat{\mathbf{q}}_t)||_2^2.$$

Then

$$\mathbb{E}_{(\mathbf{x}_t, y_t) \sim Q_t}\left[\frac{1}{T}\sum_{t=1}^{T}\ell(\mathbf{p}_t, \mathbf{q}_t)\right] - \frac{1}{T}\sum_{t=1}^{T}\ell(\mathbf{p}, \mathbf{q}_t)$$

$$= \mathbb{E}\left[\frac{1}{T}\sum_{t=1}^{T}\frac{1}{2\eta}\left(||\mathbf{p}_t - \mathbf{p}||_2^2 - ||\mathbf{p}_{t+1} - \mathbf{p}||_2^2\right) + \frac{\eta}{2}||\nabla_{\mathbf{p}}\ell(\mathbf{p}_t, \hat{\mathbf{q}}_t)||_2^2\right]$$

$$= \frac{1}{2\eta T}\left(||\mathbf{p}_1 - \mathbf{p}||_2^2 - ||\mathbf{p}_{T+1} - \mathbf{p}||_2^2\right) + \frac{\eta}{2T}\sum_{t=1}^{T}\mathbb{E}\left[||\nabla_{\mathbf{p}}\ell(\mathbf{p}_t, \hat{\mathbf{q}}_t)||_2^2\right]$$

$$\leq \frac{1}{2\eta T}||\mathbf{p}_1 - \mathbf{p}||_2^2 + \frac{\eta}{2T}\sum_{t=1}^{T}\mathbb{E}\left[||\nabla_{\mathbf{p}}\ell(\mathbf{p}_t, \hat{\mathbf{q}}_t)||_2^2\right]$$

$$\leq \frac{1}{\eta T} + \frac{\eta}{2}L^2,$$

where the last inequality take the fact that $\sup_{\mathbf{p}_1, \mathbf{p}_2 \in \Delta^{M-1}} ||\mathbf{p}_1 - \mathbf{p}_2||_2^2 \leq 2\sup_{\mathbf{p}_1 \in \Delta^{M-1}} ||\mathbf{p}_1||_2^2 = 2$. $\eta = \sqrt{\frac{2}{T}\frac{1}{L}}$ derives the bound

$$\mathbb{E}_{(\mathbf{x}_t, y_t) \sim Q_t}\left[\frac{1}{T}\sum_{t=1}^{T}\ell(\mathbf{p}_t, \mathbf{q}_t)\right] - \frac{1}{T}\sum_{t=1}^{T}\ell(\mathbf{p}, \mathbf{q}_t) \leq \sqrt{\frac{2}{T}}L.$$

As the above bound holds for any $\mathbf{p}$,

$$\mathbb{E}_{(\mathbf{x}_t, y_t) \sim Q_t}\left[\frac{1}{T}\sum_{t=1}^{T}\ell(\mathbf{p}_t, \mathbf{q}_t)\right] - \min_{\mathbf{p} \in \Delta^{M-1}}\frac{1}{T}\sum_{t=1}^{T}\ell(\mathbf{p}, \mathbf{q}_t) \leq \sqrt{\frac{2}{T}}L.$$

$\square$

**Theorem 3** (Regret bound for FTH, generalized version with $\delta \geq 0$). *For any* $\mathbf{q} \in \Delta^{M-1}$, *let* $\delta(\mathbf{q}) = ||p^*(\mathbf{q}) - \mathbf{q}||_2$ *where* $p^*(\mathbf{q}) = \arg\min_{\mathbf{p} \in \Delta^{M-1}}\ell(\mathbf{p}, \mathbf{q})$. *With Assumption 5, let* $L = \sup_{\mathbf{p}, \mathbf{q} \in \Delta^{M-1}} ||\nabla_{\mathbf{p}}\ell(\mathbf{p}; \mathbf{q})||_2 < \infty$. *Then with probability at least* $1 - 2MT^{-7}$ *over samples* $(\mathbf{x}_t, y_t) \sim Q_t$ *for* $t = 1, \ldots, T$, *we have that* $\mathcal{A}^{\mathsf{fth}}$ *satisfies:*

$$\frac{1}{T}\sum_{t=1}^{T}\ell(\mathbf{p}_t; \mathbf{q}_t) - \inf_{\mathbf{p} \in \Delta^{M-1}}\sum_{t=1}^{T}\ell(\mathbf{p}; \mathbf{q}_t) \leq 2L\frac{\ln T}{T} + 4Lc\sqrt{\frac{M\ln T}{T}} + \frac{3L}{T}\sum_{t=1}^{T}\delta\left(\frac{1}{t-1}\sum_{\tau=1}^{t-1}\mathbf{q}_\tau\right),$$

*where* $c = 2\max_{i=1,\ldots,M}\left\|\left(C_{f_0,Q_0}^{\top}\right)^{-1}\mathbf{e}_i\right\|_\infty$.

*Proof of theorem 3.* By Theorem 1 we have that $\mathbb{E}[\hat{\mathbf{q}}_t] = \mathbf{q}_t$ and $\hat{\mathbf{q}}_t$ $(t = 1, \cdots, T)$ are independent. By Hoeffding:

$$\mathbb{P}\left(\left\|\frac{1}{t}\sum_{\tau=1}^{t}\hat{\mathbf{q}}_\tau - \frac{1}{t}\sum_{\tau=1}^{t}\mathbf{q}_\tau\right\|_2 \geq \sqrt{M}\varepsilon_t\right) \leq 2M\exp\left(-\frac{2\varepsilon_t^2 t}{c^2}\right).$$

With union bound:

$$\mathbb{P}\left(\forall t \leq T, \left\|\frac{1}{t}\sum_{\tau=1}^{t}\hat{\mathbf{q}}_\tau - \frac{1}{t}\sum_{\tau=1}^{t}\mathbf{q}_\tau\right\|_2 < \sqrt{M}\varepsilon_t\right) \geq 1 - \sum_{t=1}^{T}2M\exp\left(-\frac{2\varepsilon_t^2 t}{c^2}\right).$$

Since $\mathbf{p}_t = \frac{1}{t-1} \sum_{\tau=1}^{t-1} \hat{\mathbf{q}}_\tau$ we have that $\|\mathbf{p}_t - \frac{1}{t-1} \sum_{\tau=1}^{t-1} \mathbf{q}_\tau\|_2 < \sqrt{M}\epsilon_t \; \forall t$ with probability at least $1 - \sum_{t=1}^{T} 2M \exp\left(-\frac{2\varepsilon_t^2 t}{c^2}\right)$, hence by the Lipschitz-ness of $\ell$:

$$\sum_{t=1}^{T} \ell(\mathbf{p}_t, \mathbf{q}_t) - \sum_{t=1}^{T} \ell(\mathbf{p}, \mathbf{q}_t) \leq \sum_{t=1}^{T} \ell\left(\frac{1}{t-1}\sum_{\tau=1}^{t-1}\mathbf{q}_\tau, \mathbf{q}_t\right) - \sum_{t=1}^{T} \ell(\mathbf{p}, \mathbf{q}_t) + L\sqrt{M} \cdot \sum_{t=1}^{T} \varepsilon_t. \quad (12)$$

We will first derive an upper bound for $\sum_{t=1}^{T} \ell\left(\frac{1}{t-1}\sum_{\tau=1}^{t-1}\mathbf{q}_\tau, \mathbf{q}_t\right) - \sum_{t=1}^{T} \ell(\mathbf{p}, \mathbf{q}_t)$. Recall that $p^*(\mathbf{q}) = \arg\min_{\mathbf{p}} \ell(\mathbf{p}, \mathbf{q})$ and $\delta(\mathbf{q}) = \|p^*(\mathbf{q}) - \mathbf{q}\|_2$. Then

$$\sum_{t=1}^{T} \ell\left(\frac{1}{t-1}\sum_{\tau=1}^{t-1}\mathbf{q}_\tau, \mathbf{q}_t\right) - \sum_{t=1}^{T} \ell(\mathbf{p}, \mathbf{q}_t)$$

$$\leq \sum_{t=1}^{T} \ell\left(p^*\left(\frac{1}{t-1}\sum_{\tau=1}^{t-1}\mathbf{q}_\tau\right), \mathbf{q}_t\right) - \sum_{t=1}^{T} \ell(\mathbf{p}, \mathbf{q}_t) + L \cdot \sum_{t=1}^{T} \delta\left(\frac{1}{t-1}\sum_{\tau=1}^{t-1}\mathbf{q}_\tau\right) \quad (13)$$

$$\leq \sum_{t=1}^{T} \ell\left(p^*\left(\frac{1}{t-1}\sum_{\tau=1}^{t-1}\mathbf{q}_\tau\right), \mathbf{q}_t\right) - \sum_{t=1}^{T} \ell\left(p^*\left(\frac{1}{t}\sum_{\tau=1}^{t}\mathbf{q}_\tau\right), \mathbf{q}_t\right) + L \cdot \sum_{t=1}^{T} \delta\left(\frac{1}{t-1}\sum_{\tau=1}^{t-1}\mathbf{q}_\tau\right) \quad (14)$$

$$\leq \sum_{t=1}^{T} \ell\left(\frac{1}{t-1}\sum_{\tau=1}^{t-1}\mathbf{q}_\tau, \mathbf{q}_t\right) - \sum_{t=1}^{T} \ell\left(\frac{1}{t}\sum_{\tau=1}^{t}\mathbf{q}_\tau, \mathbf{q}_t\right) + 3L \cdot \sum_{t=1}^{T} \delta\left(\frac{1}{t-1}\sum_{\tau=1}^{t-1}\mathbf{q}_\tau\right) \quad (15)$$

$$\leq \sum_{t=1}^{T} \frac{2L}{t} + 3L \cdot \sum_{t=1}^{T} \delta\left(\frac{1}{t-1}\sum_{\tau=1}^{t-1}\mathbf{q}_\tau\right) \quad (16)$$

$$\leq 2L \ln T + 3L \cdot \sum_{t=1}^{T} \delta\left(\frac{1}{t-1}\sum_{\tau=1}^{t-1}\mathbf{q}_\tau\right), \quad (17)$$

where (13) and (15) are implied by Lipschitz-ness of $\ell$, (14) holds by Lemma 2.1 in [32] and (16) holds by Lipschitz-ness of $\ell$ and the fact that

$$\left\|\frac{1}{t-1}\sum_{\tau=1}^{t-1}\mathbf{q}_\tau - \frac{1}{t}\sum_{\tau=1}^{t}\mathbf{q}_\tau\right\|_2 = \left\|\frac{\sum_{\tau=1}^{t-1}\mathbf{q}_\tau}{(t-1)t} - \frac{\mathbf{q}_t}{t}\right\|_2 \leq \frac{1}{t}\cdot\left\|\frac{\sum_{\tau=1}^{t-1}\mathbf{q}_\tau}{t-1} - \mathbf{q}_t\right\|_2 \leq \frac{2}{t}.$$

Combining with (12), with probability at least $1 - \sum_{t=1}^{T} 2M \exp\left(-\frac{2\varepsilon_t^2 t}{c^2}\right)$, we have

$$\sum_{t=1}^{T} \ell(\mathbf{p}_t, \mathbf{q}_t) - \min_{\mathbf{p}} \sum_{t=1}^{T} \ell(\mathbf{p}, \mathbf{q}_t) \leq 2L \ln T + \sum_{t=1}^{T} 3\delta\left(\frac{1}{t-1}\sum_{\tau=1}^{t-1}\mathbf{q}_\tau\right)L + L\sqrt{M} \cdot \sum_{t=1}^{T} \varepsilon_t.$$

Take $\varepsilon_t = 2c\sqrt{\frac{\ln T}{t}}$ so that $\sum_{t=1}^{T} 2M \exp\left(-\frac{2\varepsilon_t^2 t}{c^2}\right) = 2MT^{-7}$ and $\sum_{t=1}^{T} \varepsilon_t \leq 4c\sqrt{T \ln T}$. The above bound then becomes: with probability at least $1 - 2MT^{-7}$,

$$\frac{1}{T}\sum_{t=1}^{T} \ell(\mathbf{p}_t, \mathbf{q}_t) - \min_{p} \sum_{t=1}^{T} \ell(p, \mathbf{q}_t) \leq 2L\frac{\ln T}{T} + 4Lc\sqrt{\frac{M \ln T}{T}} + \frac{3L}{T}\sum_{t=1}^{T} \delta\left(\frac{1}{t-1}\sum_{\tau=1}^{t-1}\mathbf{q}_\tau\right).$$

$\square$

# B  Approximation of the gradient

In section 4.1 we defined a smooth surrogate $\ell^{\mathsf{prob}}(f; Q) := \mathbb{E}_{(\mathbf{x},y)\sim Q}[1 - P_f(\mathbf{x})[y]]$ for the expected 0-1 loss to enable direct gradient estimation. Here, we formalize the desirable properties of this surrogate loss and prove the regret bound analogue of Theorem 1 for the surrogate loss $\ell^{\mathsf{prob}}$.

**Theorem 4.** *Let $f$ be any classifier and let $Q$ be a distribution over $\mathcal{X} \times \mathcal{Y}$. Let $\ell^{\text{prob}}(f; Q) :=$ $\mathbb{E}_{(\mathbf{x},y) \sim Q}[1 - P_f(\mathbf{x})[y]]$ be the surrogate loss and let $C_{f,Q_0}^{\text{prob}}$ be its corresponding confusion matrix, with entries: $C_{f,Q_0}^{\text{prob}}[i,j] := \mathbb{E}_{\mathbf{x} \sim Q_0(\cdot|y=i)}[P_f(\mathbf{x})[j]]$. Then $\ell^{\text{prob}}$ is classification-calibrated, and is smooth in $f$ if $P_f$ is smooth in $f$. Furthermore, if $\hat{\mathbf{q}}_t$ is an unbiased estimator of $\mathbf{q}_t$ then:*

$$\ell^{\text{prob}}(f; Q_t) = \mathbb{E}_{Q_t}\left[\left\langle \mathbf{1} - \text{diag}\left(C_{f,Q_0}^{\text{prob}}\right), \hat{\mathbf{q}}_t \right\rangle\right],$$

$$\nabla_f \ell^{\text{prob}}(f; Q_t) = \mathbb{E}_{Q_t}\left[J_f^\top \hat{\mathbf{q}}_t\right],$$

*where $J_f = \frac{\partial}{\partial f}\left[\mathbf{1} - \text{diag}\left(C_{f,Q_0}^{\text{prob}}\right)\right]$.*

*Proof.* To show classification-calibratedness, we specialize the definition of [38] to our setting. That is, we need to show that for all $\mathbf{p} \in \Delta^{M-1}$:

$$\inf_{\mathbf{z} \in \Delta^{M-1} : \mathbf{p}[\arg\max_y \mathbf{z}[y]] < \max_y \mathbf{p}[y]} 1 - \langle \mathbf{p}, \mathbf{z} \rangle > \inf_{\mathbf{z} \in \Delta^{M-1}} 1 - \langle \mathbf{p}, \mathbf{z} \rangle. \qquad (18)$$

Let $p = \max_y \mathbf{p}[y]$. Since $\mathbf{z} \geq 0$, we have that $1 - \langle \mathbf{p}, \mathbf{z} \rangle \geq 1 - \langle p\mathbf{1}, \mathbf{z} \rangle = 1 - p$, which holds with equality for $\mathbf{z} = \mathbf{e}_{\arg\max_y \mathbf{p}[y]}$. Hence the RHS of Equation 18 is equal to $1 - p$. The LHS is an infimum over $\mathbf{z}$ with $\mathbf{p}[y'] < p$ where $y' = \arg\max_y \mathbf{z}[y]$. In particular, $\mathbf{z}[y'] \geq 1/M$, hence

$$1 - \langle \mathbf{p}, \mathbf{z} \rangle = 1 - \mathbf{p}[y']\mathbf{z}[y'] - \sum_{y \neq y'} \mathbf{p}[y]\mathbf{z}[y]$$

$$\geq 1 + (p - \mathbf{p}[y'])\mathbf{z}[y'] - p\mathbf{z}[y'] - \sum_{y \neq y'} p\mathbf{z}[y]$$

$$\geq 1 - p + (p - \mathbf{p}[y'])/M.$$

Taking minimum over $y'$ with $\mathbf{p}[y'] < p$ shows that Equation 18 holds and therefore $\ell^{\text{prob}}$ is classification-calibrated.

To see smoothness, observe that $\nabla_f \ell^{\text{prob}}(f; Q_t) = \mathbb{E}_{(\mathbf{x},y) \sim Q_t}[1 - \nabla_f(P_f(\mathbf{x})[y])]$. Hence smoothness of $P_f$ in $f$ implies the smoothness of $\ell^{\text{prob}}(f; Q_t)$.

Lastly, $\ell^{\text{prob}}(f; Q_t)$ can be rewritten as

$$\ell^{\text{prob}}(f; Q_t) = \mathbb{E}_{(\mathbf{x},y) \sim Q_t}[1 - P_f(\mathbf{x})[y]]$$

$$= \sum_{i=1}^M \mathbb{E}_{\mathbf{x}_t \sim Q_t(\cdot|y_t=i)}[1 - P_f(\mathbf{x})[y]] \cdot \mathbb{P}_{Q_t}(y_t = i)$$

$$= \left\langle \mathbf{1} - \text{diag}\left(C_{f,Q_0}^{\text{prob}}\right), \mathbf{q}_t \right\rangle.$$

Notice that $\text{diag}\left(C_{f,Q_0}^{\text{prob}}\right)$ is independent of $\mathbf{q}_t$. Thus $\ell^{\text{prob}}(f; Q_t)$ is a linear function of $\mathbf{q}_t$, and substituting in the unbiased estimator $\hat{\mathbf{q}}_t$ for $\mathbf{q}_t$ gives unbiased estimators for $\ell^{\text{prob}}(f; Q_t)$ and $\nabla_f \ell^{\text{prob}}(f; Q_t)$, as desired. $\qquad \square$

Similar to Assumption 2, we assume that $\ell^{\text{prob}}$ is convex in its first parameter to derive a convergence guarantee for OGD. Under this assumption, the proof of convergence is identical to that of Theorem 2. We state the assumption below and empirically verify it in the next section. Similar to Assumption 3, we also assume the Lipschitz condition for $\ell^{\text{prob}}$.

**Assumption 6** (Convexity of $\ell^{\text{prob}}$). $\forall \mathbf{q} \in \Delta^{M-1}$, $\ell^{\text{prob}}(\mathbf{p}; \mathbf{q})$ *is convex in $\mathbf{p}$.*

**Assumption 7** (Lipschitz of $\ell^{\text{prob}}$). $\sup_{\mathbf{p} \in \Delta^{M-1}, i=1,\cdots,M} \left\| \nabla_{\mathbf{p}} \ell^{\text{prob}}\left(\mathbf{p}; \left(C_{f_0,Q_0}^\top\right)^{-1} \mathbf{e}_i\right) \right\|_2$ *is finite.*

**Theorem 5** (Regret bound for OGD w.r.t. $\ell^{\text{prob}}$). *Under Assumption 6 and 7, let $L = \sup_{\mathbf{p} \in \Delta^{M-1}, i=1,\cdots,M} \left\| \nabla_{\mathbf{p}} \ell^{\text{prob}}\left(\mathbf{p}; \left(C_{f_0,Q_0}^\top\right)^{-1} \mathbf{e}_i\right) \right\|_2$. If $\eta = \sqrt{\frac{2}{T}\frac{1}{L}}$ then $\mathcal{A}_{\text{ogd}}$ w.r.t. $\ell^{\text{prob}}$ satisfies:*

$$\mathbb{E}_{(\mathbf{x}_t, y_t) \sim Q_t}\left[\frac{1}{T}\sum_{t=1}^T \ell^{\text{prob}}(\mathbf{p}_t; \mathbf{q}_t)\right] - \inf_{\mathbf{p} \in \Delta^{M-1}} \frac{1}{T}\sum_{t=1}^T \ell^{\text{prob}}(\mathbf{p}; \mathbf{q}_t) \leq \sqrt{\frac{2}{T}}L,$$

$$\text{with } L = \sup_{\mathbf{p}' \in \Delta^{M-1}} \max_{i=1,\dots,M} \left\| \nabla_{\mathbf{p}} \ell^{\text{prob}} \left( \mathbf{p}; \left( C_{f_0,Q_0}^{\top} \right)^{-1} \mathbf{e}_i \right) \Big|_{\mathbf{p}=\mathbf{p}'} \right\|_2 .$$

## C   Empirical Justification of Assumptions

In this section we provide empirical evidence for Assumptions 2-6.

### C.1   Convexity

Assumptions 2 and 6 state that $\ell(\mathbf{p}; \mathbf{q})$ and $\ell^{\text{prob}}(\mathbf{p}; \mathbf{q})$ are convex in $\mathbf{p}$. We first verify the convexity of $\ell(\mathbf{p}; \mathbf{q})$ by uniformly sampling $\mathbf{q}, \mathbf{p}_1, \mathbf{p}_2$ from $Delta_M$ and plotting the function value of $\ell(\mathbf{p}; \mathbf{q})$ for $\mathbf{p} \in [\mathbf{p}_1, \mathbf{p}_2]$.

The left plot of Figure 1(a) shows the function value of $h(t) = \ell((1-t) \cdot \mathbf{p}_1 + t \cdot \mathbf{p}_2; \mathbf{q})$ for $t \in [0, 1]$. It can be seen that all 5 curves are approximately convex in $t$. In the right histogram plot, we evaluate $h(0.5) - \frac{1}{2}(h(0) + h(1))$ for randomly chosen tuples $\mathbf{q}, \mathbf{p}_1, \mathbf{p}_2$, which should be non-positive if Assumption 2 holds. Indeed, among 10,000 random samples, $h(0.5) - \frac{1}{2}(h(0) + h(1)) \leq 0$ holds true 99.5% of the time. For the remaining 0.5% with positive difference, the deviation from 0 is quite small, which is likely due to the estimation error for $\ell$. These empirical observations validate the assumption that the loss function $\ell(\mathbf{p}; \mathbf{q})$ is convex in $\mathbf{q}$.

We can observe similar trends for different (dataset, model) pairs in (CIFAR10, ResNet50), (SVHN, ResNet18) and (SVHN, ResNet50), as shown in Figure 1(b-d). Figure 2 shows the same trend for the various datasets and models for the surrogate loss $\ell^{\text{prob}}(\mathbf{p}; \mathbf{q})$.

### C.2   Symmetric optimality

To empirically validate the *symmetric optimality* assumption (Assumption 4), we measure the $L_2$ distance $||\mathbf{q} - \arg\max_{\mathbf{p} \in \Delta_M} \ell(\mathbf{p}; \mathbf{q})||_2$ for 1000 randomly sampled $\mathbf{q}$. We evaluate both un-calibrated and calibrated base classifiers $f_0$, where calibration is done using temperature scaling [12]. Same as above, we evaluate on both CIFAR10 and SVHN using ResNet18 and ResNet50 base classifiers. The optimal re-weight factor $\mathbf{p} := \arg\max_{\mathbf{p} \in \Delta^{M-1}} \ell(\mathbf{p}; \mathbf{q})$ is computed by optimizing $\ell(\mathbf{p}; \mathbf{q})$ with gradient ascent using gradient estimates obtained from Algorithm 3.

Figure 3 shows the histogram of the $L_2$ distances $||\mathbf{q} - \arg\max_{\mathbf{p} \in \Delta_M} \ell(\mathbf{p}; \mathbf{q})||_2$ for different samples of $\mathbf{q}$. The left plot in each subfigure shows the result for a well-calibrated $f_0$, where the distance is skewed towards 0 with more than half of the samples having distance smaller than 0.005. In comparison, the right plot for un-calibrated $f_0$ has a much higher density for larger values, which shows that a well-calibrated classifier better satisfies Assumption 4.

## D   Dataset Processing and Training Set-up

**ArXiv dataset processing.**   We select papers from the Computer Science domain and use the first category as the true label $y$. We specifically consider the 23 most populated categories, which are cs.NE, cs.SE, cs.LO, cs.CY, cs.CV, cs.SI, cs.AI, cs.CR, cs.SY, cs.PL, cs.CL, cs.IR, cs.RO, cs.DS, cs.NI, cs.CC, cs.GT, cs.LG, cs.IT, cs.DM, cs.HC, cs.DB, cs.DC.

For the feature vector $\mathbf{x}$, we compute the tf-idf vector of each paper's abstract after removing words that appear in less than 30 papers among all papers in the dataset. We further remove papers whose numbers of words after the previous filtering step is smaller than 20.

**Training set-up for the base model $f_0$.**   For the experiments on CIFAR10 under simulated shift, we train the base ResNet18 classifier $f_0$ for 150 epochs using Adam with batch size as 128, and drop learning rate twice at $1/2$ and $3/4$ of total training epochs. For the experiments on ArXiv, we train a multinomial regression model $f_0$ with L2 regularization. The L2 regularization coefficient in the loss function is selected as $10^{-6}$, which achieves the best validation accuracy among the choices in $\{10^{-5}, 10^{-6}, 10^{-7}, 10^{-8}\}$.

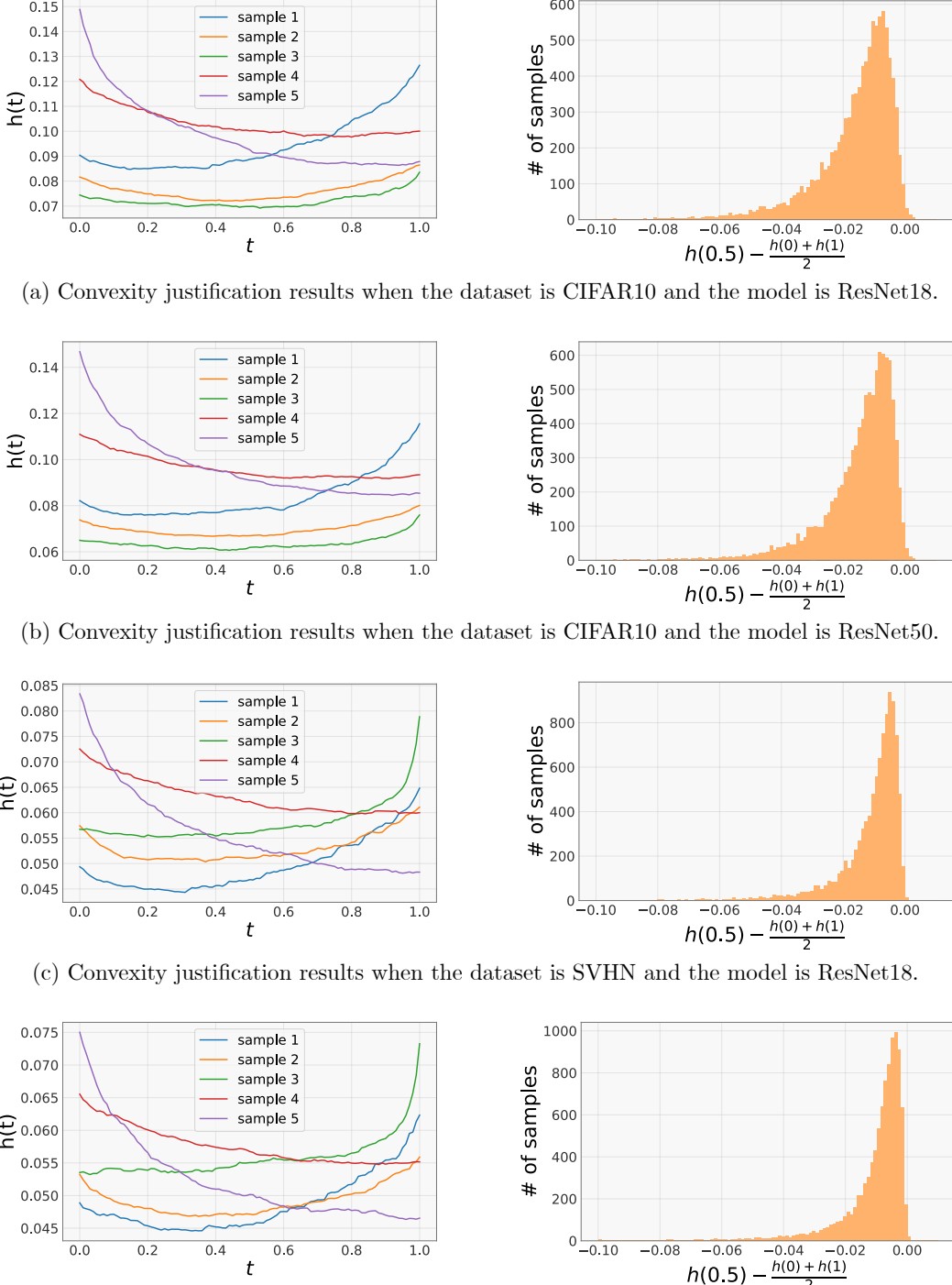

(a) Convexity justification results when the dataset is CIFAR10 and the model is ResNet18.

(b) Convexity justification results when the dataset is CIFAR10 and the model is ResNet50.

(c) Convexity justification results when the dataset is SVHN and the model is ResNet18.

(d) Convexity justification results when the dataset is SVHN and the model is ResNet50.

Figure 1: Empirical justification for the convexity assumption for $\ell(\mathbf{p}; \mathbf{q})$. See text for details.

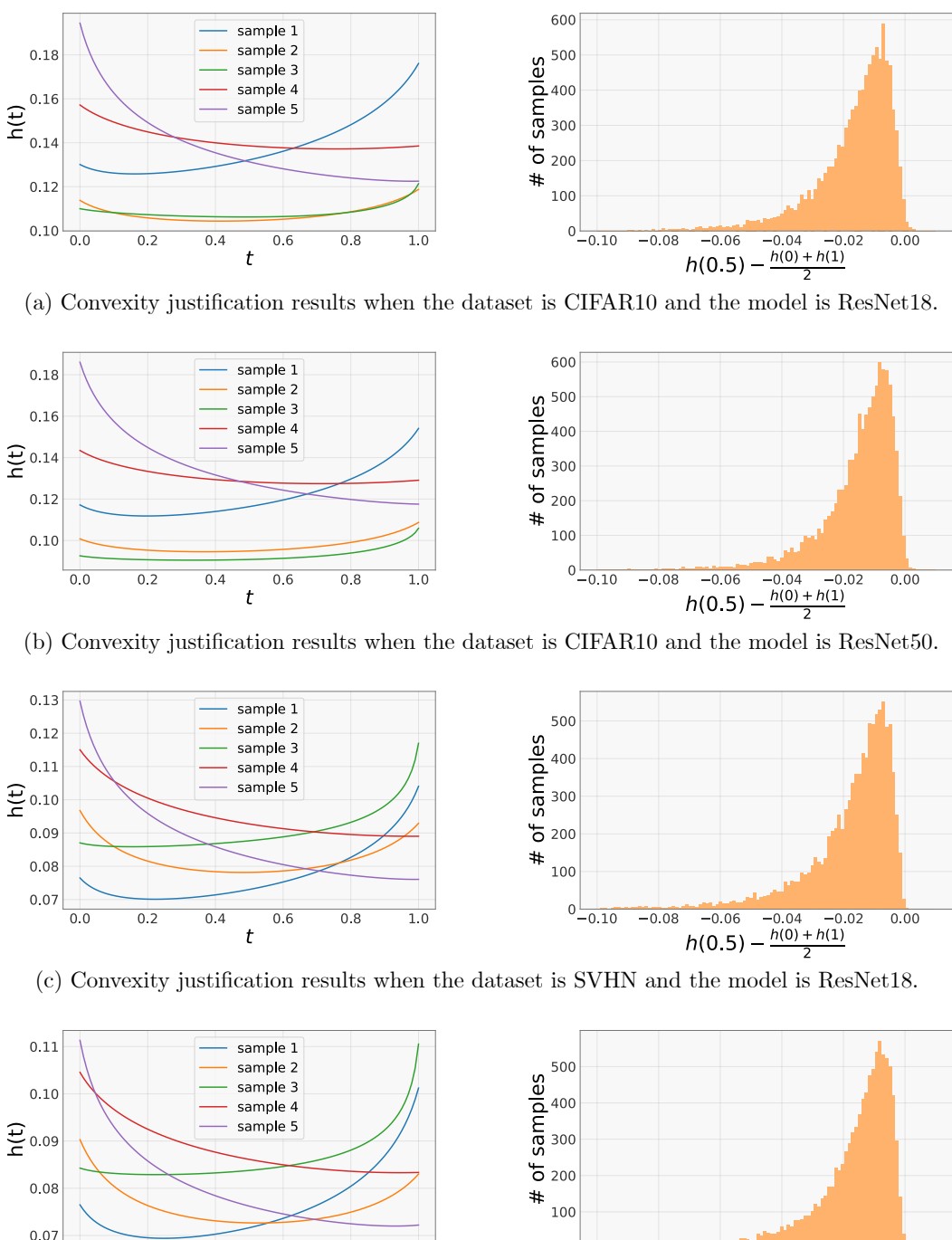

(a) Convexity justification results when the dataset is CIFAR10 and the model is ResNet18.

(b) Convexity justification results when the dataset is CIFAR10 and the model is ResNet50.

(c) Convexity justification results when the dataset is SVHN and the model is ResNet18.

(d) Convexity justification results when the dataset is SVHN and the model is ResNet50.

Figure 2: Empirical justification for the convexity assumption for $\ell^{\mathsf{prob}}(\mathbf{p}; \mathbf{q})$. See text for details.

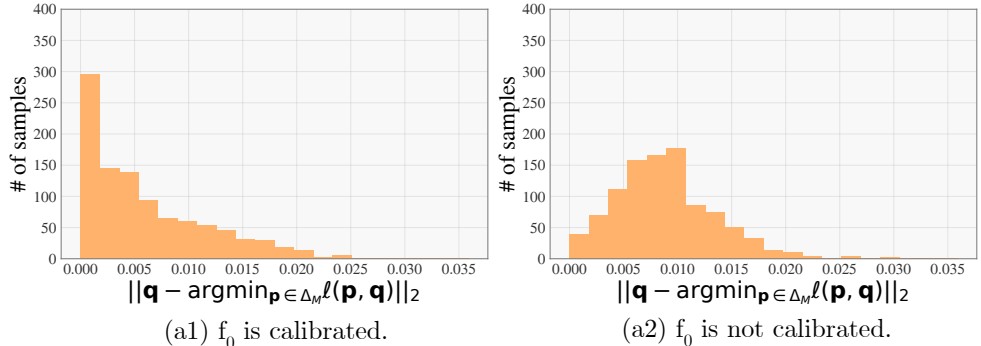

(a) Convexity justification results when the dataset is CIFAR10 and the model is ResNet18.

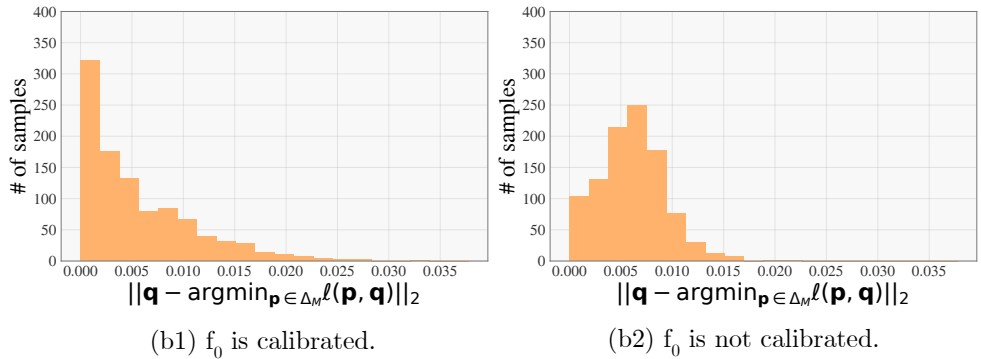

(b) Convexity justification results when the dataset is CIFAR10 and the model is ResNet50.

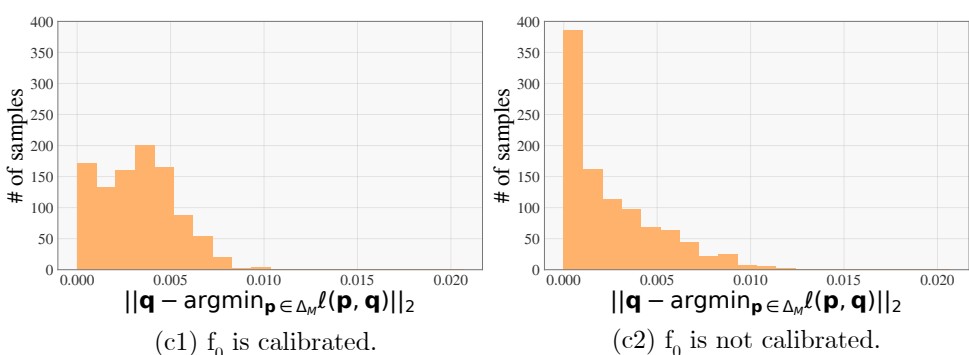

(c) Convexity justification results when the dataset is SVHN and the model is ResNet18.

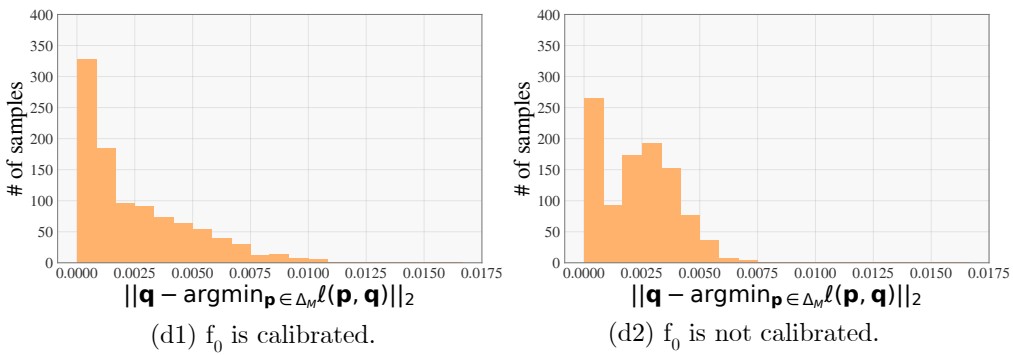

(d) Convexity justification results when the dataset is SVHN and the model is ResNet50.

Figure 3: Empirical justification for the symmetric optimality assumption. See text for details.