# OpenReview forum: "Online Adaptation to Label Distribution Shift"
_NeurIPS.cc/2021/Conference — NeurIPS 2021 Poster_

### Official Review · Reviewer_EUMs · 2021-07-12

**Rating:** 7
**Confidence:** 3

**Summary:**

The paper proposes methods for adapting models in an online manner when the label distribution P(y) is continually changing but the class conditional distribution p(x|y) is fixed, and the labels are not available. It shows a surprisingly elegant argument that the expected 0-1 loss and its gradient can be computed without the labels, and hence the models can be adapted in an online manner.
Through experiments on different simulated label shifts on CIFAR data, and on a real dataset from arXiv, it shows that one of the proposed methods OGD is simple and performant in adapting to online label distribution shifts.


**Ethical Concerns:**


The authors can address the societal impact of this work. For example, if these methods are used in social systems (say, to model credit default probabilities, risk propensity, dangerous behavior etc.), it is critical to use covariates where the class conditional probabilities do not change. For example, typical covariates in these models like demographics, behavior all change with time, and even the class conditional probabilities can change over time. So, the authors can mention in the paper for the reader/user of this method to be cautious when applying this method in practice.

**Ethics Review Area:**

["I don’t know"]

**Limitations And Societal Impact:**

My main concern is how valid of the assumption that the class-conditional distribution p(x|y) does not change, in real world situations. It is likely true in the one real dataset they tested viz. arxiv; however, how general is this assumption valid in practice? I am less clear that the same assumption holds for example in the healthcare (flu vs. hay fever) mentioned in the paper. For example, even between different episodes of flu, the variants are slightly different (although generally under the class "flu") and these differences in class can translate into differences in the covariates.


**Main Review:**

Originality: The paper is one of the few focusing on model adaptation for label distribution shifts in an online setting.  Through a simple and elegant argument (equation 3 and Theorem 1) it shows how the model can be adapted in this online setting. Although simple, this appears to be the first paper to identify this connection, and exploit it to update models in this online manner.

Quality: The paper is of good quality - the arguments seem sound, theorems and assumptions are explained and proved, and bounds are shown clearly. The experiments on both simulated label shifts on CIFAR data and a real dataset from arXiv are sound. The results look plausible and convincing.  The paper presents good details on implementation - for example using both finite difference and surrogate losses to estimate the loss gradient, and carries forward these details through the experiments. The intuitive arguments, mathematical arguments, theorems and proofs and experiments appear rigorous.

Clarity: The paper is written well, with clearly stated assumptions, theorems, frameworks, algorithms, experiment methodologies and results.  I have not checked the proofs in the appendix.

Significance: Label distribution shifts is a common problem in real world applications. It is not clear how well the assumption that P(x|y) holds in practice, but under this assumption the proposed methods - and especially the simple OGD method that is performant is interesting, useful and significant. I can see this method becoming popular in applied machine learning in practice,  due to its simplicity and performance.



**Time Spent Reviewing:**

3.5 hours

---

> ### Author Response · Authors · 2021-08-10
> **Official Comment by Paper3568 Authors**
>
> Thank you for your insightful comments and positive feedback on our work!
>
> **Label shift assumption in real world situations**: In our view, the purpose of studying label shift is to restrict the general problem of domain shift to a more tractable setting that admits principled algorithms. As you pointed out, the assumption is unlikely to hold exactly in practice, but even approximately satisfying the label shift assumption can be sufficient for label shift adaptation algorithms to perform well. For the arXiv dataset, we verified that the label shift assumption indeed holds approximately; please refer to our response to reviewer BpAt.

---

### Official Review · Reviewer_qmA9 · 2021-07-13

**Rating:** 6
**Confidence:** 3

**Summary:**

This paper studies how to learn with target shift in an online fashion, especially when the label distribution could shift over time. For the target shift problem, the key challenge is how to estimate the probability of each label in testing time (or at every iteration in the online fashion), and then one can adapt the original classifier ($f_0$) to make good predictions. To address this issue, based on the previous study [40], authors first approximate the true label probability $q_t$ by its unbiased estimator, and then use the unbiased estimators to update the next guess $p_{t+1}$ for the true label probability $q_{t+1}$ with several classical online algorithms. Theoretical analysis shows that the proposed algorithms enjoy sublinear regret bounds in expectation. Experiments also validate the effectiveness of the methods.



**Ethical Concerns:**

No ethical concerns.

**Limitations And Societal Impact:**

As mentioned above, the performance measure (the regret) used in the theoretical analysis might be too weak for the case where the label probability could change over time. The time-varying comparator could be a better choice.

**Main Review:**

Novelty: The problem studied in the paper is novel and realistic. To my knowledge, there seem little works considering how to handle the changing label probability in an online fashion. Although the proposed methods are largely established on the previous studies (i.e. the unbiased estimator developed in [40] and the use of the classical online algorithms), I found it an interesting solution to the problem.

Quality: My main concern about the paper is the mismatch between the initial goal and the performance measure in Theorem 2. Theorem 2 shows that the proposed online algorithm can predict as well as the single best classifier in the hypothesis space. However, since the underlying label probability is changing over time, it is unconvincing that a fixed model could perform well over all iterations. Maybe a time-varying comparator could be a better choice.

Besides, the proposed algorithms seem that heavily rely on the quality of the initial models. Although the estimator is unbiased for $q_t$, its asymptotic behavior could be influenced by $f_0$. I think it would be better if there are some analyses or experiments to illustrate the influence on the quality of the initial model.

Clarity: The paper is clearly written and well organized.

Overall, I think the paper falls in the borderline case but has the merits that outweigh the flaws.



**Time Spent Reviewing:**

5-6 hours

---

> ### Author Response · Authors · 2021-08-10
> **Official Comment by Paper3568 Authors**
>
> Thank you for your insightful comments and positive feedback on our work!
>
> **Performance measure in Theorem 2**: In practical settings, the test distribution may change in unpredictable manners and it is infeasible to estimate an accuracy metric based on the unknown optimal predictor at each time step. The optimal fixed classifier (OFC), on the other hand, can be computed in hindsight to evaluate the regret metric.
> We emphasize that OFC is actually a strong benchmark because it requires knowledge of the entire sequence of test distributions $Q_t$ ahead of time, which is prohibited under the online setting. Moreover, the empirical performance of OFC is surprisingly good as shown in Tables 1 and 2, matching or outperforming FTH in most cases. Comparison against such oracle predictors is also standard in the online learning literature where no assumption is placed on the input chosen at each time step.
>
> **Influence of the initial model**: Following your suggestion, we ran an additional experiment using a ResNet50 base classifier for CIFAR-10 (which achieves a lower test error than the ResNet18 model used in the paper) and followed the same simulation set-up in section 5.2 of the paper. The results are:
>
> Base Classifier      		12.37	10.79	10.80	10.80	10.80	10.97	11.24
>
> Opt. Fixed Classifier      	7.38	9.41	9.42	9.41	9.41	9.52	9.39
>
> FTH      			7.42	9.24	9.44	9.42	9.32	9.62	9.49
>
> OGD (surrogate loss)      	7.48	9.00	9.53	9.54	9.41	8.55	8.00
>
> The columns correspond to the 7 simulated shift processes in Table 1. We can make similar observations about OGD and FTH for the ResNet50 base model. Moreover, because the ResNet50 base classifier is better than ResNet18 on the training set, applying online adaptation algorithms to it also gives better results on the test distribution. Extrapolating from this observation, we believe that it is important to train $f_0$ to first perform well on the training distribution before the online phase begins. We will include these results in the final version.

---

> > ### Comment · Reviewer_qmA9 · 2021-08-26
> > **Thank you for the response**
> >
> > Thank you for the response. I appreciate your effort to address my concerns and the additional experiment.
> >
> > As for the first question, I understand that OFC is an infeasible algorithm in the online learning setting since it always performs the best single model in hindsight. But, I am still not very convinced that it is a strong benchmark for the distribution change problem. For example, considering the case where underlying distribution changes abruptly at a certain point $T_0$, I am not sure whether a single model can perform well over all iterations. If the entire sequence of $Q_t$ is known, perhaps a better benchmark is to train two models for the period $t<T_0$ and $t>T_0$, respectively. So, I think it would be nice to enable the online algorithms to compare such a kind of benchmark.
> >
> > Having said that, I still find the paper provides an interesting method to handle the label shift problem in the online setting by using the unbiased estimator. So, I would like to stay with my score.

---

> > > ### Author Response · Authors · 2021-08-26
> > > **Re: Comparison to an adaptive baseline**
> > >
> > > We agree that it may be important to empirically compare to a more powerful adaptive baseline. Our original intention to compare to OFC in the experiment section is to show that the online adaptation algorithms can indeed achieve a low regret as predicted by Theorems 2 and 3. However, empirically they may outperform the OFC baseline since it is a fixed classifier.
> > >
> > > In this regard, reviewer BpAt proposed an interesting baseline of OGD but with known test labels, which is in fact the classical OGD in the online learning setting. This is also an infeasible classifier, but unlike OFC, it is adapting to the changing test distribution. Our observation is that OGD using Theorem 1 to estimate the stochastic gradient performs almost as well as OGD with known test label, which shows that the reduction framework is near-optimal for OGD. We plan on expanding this analysis for the final version. Does this address your concern?

---

> > > > ### Comment · Reviewer_qmA9 · 2021-08-26
> > > > **Re: Re: Comparison to an adaptive baseline**
> > > >
> > > > Thank you for the response and the new empirical result.
> > > >
> > > > It is nice to show that the proposed methods are comparable to classical OGD fed with the ground-truth label. But, the classical OGD is still guaranteed to compare with a fixed benchmark, i.e. the best model in the hypothesis, with sublinear regret [1, Theorem 3.1]. So, though the empirical result is convincing enough to show the reduction is near-optimal for OGD, it might not be strong evidence to show that the single best model is a proper benchmark in changing environments.
> > > >
> > > > Actually, in the online learning literature, there are studies on dynamic regret [2,3], which compares the performance of the learning algorithm to time-varying comparators. Perhaps an online algorithm enjoying such kinds of guarantees would better match the motivation of the paper.
> > > >
> > > > [1] Elad Hazan. Introduction to Online Convex Optimization. Found. Trends Optim. 2(3-4): 157-325 (2016)
> > > >
> > > > [2] Ali Jadbabaie, Alexander Rakhlin, Shahin Shahrampour, Karthik Sridharan. Online Optimization: Competing with Dynamic Comparators. AISTATS'15
> > > >
> > > > [3] L. Zhang, S. Lu, and Z.-H. Zhou. Adaptive Online Learning in Dynamic Environments. NeurIPS'18.

---

> > > > > ### Author Response · Authors · 2021-08-27
> > > > > **Promising future direction**
> > > > >
> > > > > Thank you for the reference to online learning with dynamic regret. We agree that it is possible to define the evaluation metric for online adaptation in terms of dynamic regret and analyze existing algorithms against a dynamic benchmark, which can serve as a promising direction for future work. We will include a discussion of this aspect in the paper.

---

### Official Review · Reviewer_BpAt · 2021-07-16

**Rating:** 6
**Confidence:** 3

**Summary:**

This paper proposes a framework for studying problems in which the test time distribution of labels can change continuously, and the model can adapt in an online fashion but is not provided any ground truth test labels. Theoretical results demonstrate that, when there is label shift but no concept shift, an unbiased estimate of the expected 0-1 loss and its gradient can be computed even without test labels. Using this insight, two online learning methods are proposed for this setting that outperform non adaptive base and oracle models in CIFAR-10 and arXiv classification problems.

**Limitations And Societal Impact:**

The authors are encouraged to include additional discussion of the limitations and potential societal impacts of their work.

**Main Review:**

Originality
---
To the best of my knowledge, this paper proposes a novel problem setting, produces novel theoretical results, and provides novel empirical methods. However, I am not an expert in the topics of online learning and label distribution shift.


Quality
---
This paper is of relatively high quality. I did not carefully check the proofs for correctness, but the theory appears generally sound and provides a clear direction for devising the practical methods. It is nice that the paper provides some empirical evidence as to the reasonableness of the assumptions. One additional assumption that would be good to test empirically is the core assumption of no concept shift, i.e., p ( x | y ) remaining constant. This is, of course, hard to verify or measure, but the authors could study a controlled setting in which there is control over how much p ( x | y ) changes, e.g., ImageNet superclass classification where the subclasses change over time, in order to better understand how crucial this assumption is in practice. This would also provide another set of experiments, which would help to strengthen the paper but, in my opinion, is not by itself critical.

Additionally, another natural point of comparison would be an even stronger oracle that is allowed to adapt with knowledge of the test labels, i.e., a "standard" online learner. This would help the reader quantify how much is lost from not having access to any labels at test time.


Clarity
---
The paper is generally well written and structured. There seem to be enough details that one could reasonably expect to be able to reproduce the methods and results. In the experiments, a few additional details would be helpful for intuition:

- Plots of sliding window average model accuracies (or some related quality) over the course of testing, in order to better understand how the models actually adapt and how rapidly this process occurs.
- A visualization or description of the label distributions in the arXiv dataset, perhaps split up by year, in order to better understand the problem.
- Some additional clarifying details about the arXiv problem setup. For example, are the data splits done by year, such that the validation set is also a different set of years (and thus also have label shift) from the training set? This seems like it could pose issues regarding model calibration and confusion matrix computation. This also appears to violate the detail in L80 about Q_0 = Q_train, so some commentary about the necessity of that assumption would also be helpful.


Significance
---
In general, this work should be of interest to researchers studying online learning, label distribution shift, and adaptation. Furthermore, practitioners working on problems which naturally exhibit this type of test setting should also find these methods interesting. However, I cannot claim this with absolute certainty -- the paper mentions the example of medical diagnosis a few times, but there are no experiments to verify this intuition in this setting, so additional work would be needed to verify the methods' applicability to other real world problems.

**Time Spent Reviewing:**

3

---

> ### Author Response · Authors · 2021-08-10
> **Official Comment by Paper3568 Authors**
>
> Thank you for your insightful comments and positive feedback on our work!
>
> **Empirical verification for the no concept shift assumption**: The primary use for the label shift assumption is the computation of the confusion matrix $C_{f, Q_t}$ for time step $t>0$, which is made possible by the assumption that $p(x | y)$ remains constant, hence $C_{f, Q_t} = C_{f, Q_0}$. We empirically tested this hypothesis for the arXiv dataset by evaluating the confusion matrix $C_{f, Q}$ where $Q$ is the empirical distribution of data across all time steps $t>0$. The MSE between $C_{f, Q_0}$ and $C_{f, Q}$ is 0.0002, which suggests that there is negligible concept shift across time. We will include this analysis in the final version of the paper if accepted.
>
> **Comparison to the oracle with test labels**: Great suggestion! Access to the test label means that the 0-1 loss on test distribution $Q_t$ can be unbiasedly estimated using the stochastic loss on the test sample $(x_t, y_t)$. Thus, a standard online learner such as OGD that utilizes this unbiased estimator has the same convergence rate as what we showed in Theorem 2. We implemented this OGD learner using gradient of the surrogate loss for the 7 simulated shift processes in Table 1:
>
> OGD (unknown test label)	7.78    9.75    10.24    10.21    10.05    8.92    8.50
>
> OGD (known test label)	7.67    9.71    10.10    10.14    10.02    8.89    8.47
>
> This result shows that applying OGD with gradient estimated by Theorem 1 performs just as well as knowing the true label at test time! We will include a more thorough analysis of this result in the final version if accepted.
>
> **Visualization of results and dataset description**: Thanks for those suggestions. We will include it in the camera ready if accepted.
>
> **Assumptions for $Q_0$ and $Q_{train}$**: Thanks for pointing this out! The training and validation sets for arXiv are split by year as well. This is okay for estimating the confusion matrix under the label shift assumption, but may affect the result of calibration. We reran the experiment but sample the calibration data from the same distribution as the training set. The result is indeed very similar to what we presented in Table 2:
>
> Base	Opt. Fixed	FTH	FTFWH(w=100)	FTFWH(w=1000)	FTFWH(w=10000)	OGD(surr. loss)
>
> 27.44	26.06		26.16	30.74		26.66		26.41			26.18

---

> > ### Comment · Reviewer_BpAt · 2021-08-26
> > **Thanks for your response**
> >
> > Thanks for your response, I appreciate the clarifications and additional experiments.The comparison to the oracle is nice -- it looks to me like the oracle may slightly outperform the proposed method, which would not be surprising since it has access to labels. It is hard for me to judge whether or not your claim that the proposed method "performs just as well" as the oracle is completely justified given the results.
> >
> > Regardless, I am inclined to stick with my original recommendation, and I believe that this work meets the bar for publication.

---

### Official Review · Reviewer_r9Tm · 2021-07-16

**Rating:** 6
**Confidence:** 4

**Summary:**

This paper proposes FTH and OGD methods against the online label shift problem, and the experimental results demonstrate that the proposed methods are effective and robust to various online label shift scenarios.

**Limitations And Societal Impact:**

See the section "Main Review"

**Main Review:**

As far as I know, this is the first paper considering the online label shift problem which is practical and important for many real-world machine learning problems. The authors give a detailed analysis targeting the online label shift problem and propose solutions for this problem. The well-designed experiments also demonstrate the effectiveness of the proposed methods. Although the result shown in this paper is satisfactory and the analysis is thorough, I have some questions in some detail.

1. Is the proposed method effective for datasets with large label space? The current label shift method often fails to deal with the large label space dataset, such as Cifar100, Imagenet. According to Table 2 in the main paper, the proposed methods seems to improve marginally in the dataset with  23 categories but such a dataset with an even larger label space is common in real-world problems.

2. Is the methods still effective/better if we only fine-tune the final layer of the classifier (the layer before softmax)?  According to the experimental results of the related paper [1], label shift is arisen by the last layer of the classifier if the classifier is trained on the balanced training dataset.

3. I fail to find the difference between "Label distribution shift" and the existing "Label Shift", and so it is unnecessary to introduce a new concept into the current research area. It is better to use a new title related to "Label Shift".

[1] Guo, J., Gong, M., Liu, T., Zhang, K., & Tao, D. (2020, November). LTF: A Label Transformation Framework for Correcting Label Shift. In International Conference on Machine Learning (pp. 3843-3853). PMLR

**Time Spent Reviewing:**

4

---

> ### Author Response · Authors · 2021-08-10
> **Official Comment by Paper3568 Authors**
>
> Thank you for your insightful comments and positive feedback on our work!
>
> **Large label space**: Similar to prior work on black-box shift estimation (BBSE; https://arxiv.org/pdf/1802.03916.pdf), one requirement for our framework is that the confusion matrix, which has size MxM where M is the number of labels, can be estimated accurately using the holdout set $D_0$. A large output space will require a larger holdout set to accurately estimate the confusion matrix. This is not a severe limitation since training data is assumed to be abundant as it does not need to reflect the test label distribution. In addition, it may be possible to apply smoothing to the confusion matrix to reduce estimation variance for small holdout sets; we consider this extension a promising direction for future work.
>
> **Fine-tuning the final layer**: Our framework indeed allows an extension to update the model parameters as well. For example, an unbiased estimate of the gradient with respect to the final layer can be computed to enable OGD to learn the optimal final layer parameters at test time.
>
> We implemented this algorithm for the CIFAR-10 experiment (fine-tuning the last layer of a trained ResNet18 classifier using the optimal learning rate) and the result is as follows:
>
> Hypothesis space - reweight		7.63	9.57	10.16	10.24	8.04	7.81	7.65
>
> Hypothesis space - the final layer	7.87	9.59	10.3	9.33	7.86	7.86	7.78
>
> The different columns correspond to the 7 simulated shift processes in Table 1. For this experiment, fine-tuning the final layer resulted in comparable performance to the reweighted model except for Periodic Shift ($T_p=1000$), where the error dropped from 10.24 to 9.33. Motivated by this result, we believe that extending our framework to a more general hypothesis set can be a promising future direction.
>
> **“Label Shift” vs “Label Distribution Shift”**: Thanks for the suggestions. We will fix it if possible.

---

> > ### Comment · Reviewer_r9Tm · 2021-08-19
> > **Re: Rebuttal**
> >
> > Thank you very much for the response. I appreciate the effort that the authors put into addressing my questions.
> >
> > 1. Thanks for the response and the experimental results are impressive. Optimizing only the final layer of the classifier can largely reduce the training time of online learning algorithms but does not harms the final results. Considering the importance of training efficiency in the context of online learning, it would be better to incorporate the experimental results and the discussion with [1] into the revised paper, which may inspire subsequent works.
> >
> > 2. Considering the numerous previous works [1,2,3,4] on "label shift",  it is unnecessary to introduce a new concept with the same setting.
> >
> > [1] Guo, J., Gong, M., Liu, T., Zhang, K., & Tao, D. (2020, November). LTF: A Label Transformation Framework for Correcting Label Shift. In International Conference on Machine Learning (pp. 3843-3853). PMLR
> >
> > [2] Lipton, Z., Wang, Y. X., & Smola, A. (2018, July). Detecting and correcting for label shift with black box predictors. In International conference on machine learning (pp. 3122-3130). PMLR.
> >
> > [3] Azizzadenesheli, K., Liu, A., Yang, F., & Anandkumar, A. (2019). Regularized learning for domain adaptation under label shifts. arXiv preprint arXiv:1903.09734.
> >
> > [4] Zhang, K., Schölkopf, B., Muandet, K., & Wang, Z. (2013, May). Domain adaptation under target and conditional shift. In International Conference on Machine Learning (pp. 819-827). PMLR.

---

> > > ### Author Response · Authors · 2021-08-19
> > > **Thank you for the suggestions**
> > >
> > > 1. We will incorporate these results on finetuning the final layer in the final version if accepted.
> > > 2. We agree that "label shift" is the more established term in this area, and will change the paper title if it's still permitted to do so for the final version.

---

### Decision · Program_Chairs · 2021-09-28

**Decision:**

Accept (Poster)

**Comment:**

The paper studies the problem of label distribution shifts in the online setting, and proposes two algorithms which guarantee low regret without the need to ever observing the true labels or losses on the test data sequence.

The paper received uniformly positive reviews. The reviewers appreciated:
- novel setting (first study of label distribution shift in the online setting), potentially important for real-world machine learning problems
- original and novel theoretical results
- comprehensive experimental study
- clarity of presentation.

Given the above, I also recommend the paper to be accepted.

**Consistency Experiment:**

NeurIPS has a long history of experimentation. In 2014, NeurIPS ran an experiment in which 10% of submissions were reviewed by two independent committees to quantify the randomness in the review process. This year, we repeated a variant of this experiment to see how the quality of the review process has changed over time.  This paper was part of the experiment and was therefore assigned to two committees (consisting of reviewers, an Area Chair, and a Senior Area Chair) that reached independent decisions.  If both committees made the same recommendation, this recommendation was followed. If a single committee recommended acceptance, the paper was accepted (with the exception of a few cases in which the other committee identified what we considered a fatal flaw, e.g., an error in a key result).

Both committees reached the same decision: **Accept (Poster)**

The other committee assigned to the paper recommended **Accept (Poster)**.  You can find the other set of reviews, along with any follow up discussion with the authors here:
https://openreview.net/forum?id=j7YA-y0P3-